# Mapping soil hydraulic properties using random forest based pedotransfer functions and geostatistics

Brigitta Szabó[1,2], Gábor Szatmári[1], Katalin Takács[1], Annamária Laborczi[1], András Makó[1,2], Kálmán Rajkai[1], László Pásztor[1]

[1]Institute for Soil Sciences and Agricultural Chemistry, Centre for Agricultural Research, Hungarian Academy of Sciences, Herman Ottó út 15, 1022 Budapest, Hungary
[2]Georgikon Faculty, University of Pannonia, Deák Ferenc u. 16, 8360 Keszthely, Hungary

*Correspondence to*: Gábor Szatmári (szatmari@rissac.hu)

**Abstract.** Spatial 3D information on soil hydraulic properties for areas larger than plot scale are usually derived using indirect methods such as pedotransfer functions (PTFs) due to the lack of measured information on them. PTFs describe the relationship between the desired soil hydraulic parameter and easily available soil properties based on a soil hydraulic reference dataset. Soil hydraulic properties of a catchment or region can be calculated by applying PTFs on available soil maps. Our aim was to analyse the performance of (i) indirect (using PTFs) and (ii) direct (geostatistical) mapping methods to derive 3D soil hydraulic properties. The study was performed on the Balaton catchment area in Hungary, where density of measured soil hydraulic data fulfils the requirements of geostatistical methods. Maps of saturated water content (0 cm matric potential), field capacity (-330 cm matric potential) and wilting point (-15000 cm matric potential) for 0-30, 30-60 and 60-90 cm soil depth were prepared. PTFs were derived using the random forest method on the whole Hungarian soil hydraulic dataset, which includes soil chemical, physical, taxonomical and hydraulic properties of some 12,000 samples complemented with information on topography, climate, parent material, vegetation and land use. As a direct, thus geostatistical method random forest combined with kriging (RFK) was applied to 359 soil profiles located in the Balaton catchment area. There were no significant differences between the direct and indirect methods in six out of nine maps having root mean squared error values between 0.052 and 0.074 cm$^3$ cm$^{-3}$, which is in accordance with the internationally accepted performance of hydraulic PTFs. The PTFs based mapping method performed significantly better than the RFK for the saturated water content at 30-60 and 60-90 cm soil depth, in the case of wilting point the RFK outperformed the PTFs at 60-90 cm depth. Differences between the PTF based and RFK mapped values are less than 0.025 cm$^3$ cm$^{-3}$ for 65-86 % of the catchment. In RFK, uncertainty of input environmental covariate layers is less influential on the mapped values which is preferable. In the PTFs based method the uncertainty of mapping soil hydraulic properties is less computational intensive. Detailed comparisons of maps derived from the PTF based method and the RFK are presented in this paper.

## 1 Introduction

Providing information on soil hydraulic properties is desired for many environmental modelling studies (Van Looy et al., 2017). Most often, measured information on soil water retention or hydraulic conductivity is not available for environmental modelling either at regional or continental scale. Analyses on the prediction of soil hydraulic properties were started extensively in the 1980s (Ahuja et al., 1985; Pachepsky et al., 1982; Rawls and Brakensiek, 1982; Saxton et al., 1986; Vereecken et al., 1989) and are continuously updated to increase the performance of predictions (pedotransfer functions - PTFs) when newer statistical methods and/or new data become available. Latest works include among others McNeill et al. (2018); Román Dobarco et al. (2019); Zhang and Schaap (2017).

Tree based machine learning algorithms have been found to be efficient tools in general for predicting purposes (Caruana et al., 2008; Caruana and Niculescu-Mizil, 2006; Olson et al., 2017), especially gradient tree boosting and random forest. These methods are used to derive ensembles of trees, providing predictions of several individual trees with built in randomization. Tree type algorithms provide mean values of groups that can be statistically differentiated, called terminal nodes (Breiman, 2001). Due to this way of providing estimations, these methods do not derive any extraordinary values, therefore predictions will always be reasonable if training data is appropriately cleaned. For the same reason it decreases variability as well, extreme values are smoothed out (Hengl et al., 2018b).

Ensemble predictions can be derived not only from a single method, which consist of several models through bagging or boosting of e.g. decision tree, or support vector machine, or neural network algorithms, but can consist of different models and is derived from the average of all. It has been shown that often but not always, the more models are combined for the prediction the more accurate the results are (Baker and Ellison, 2008; Cichota et al., 2013; Nussbaum et al., 2018; Wu et al., 2018). Although the significancy of improvement is often not tested. Hengl et al. (2017) also used merged ensemble predictions by calculating the weighted average of two machine learning algorithms to decrease influence of model overfitting. Although from the application point of view it is important to avoid increasing the complexity and size of the prediction model if there is no significant improvement in performance. Accuracy, interpretability and computation power required to use the prediction algorithm have to be optimized at the same time for allowing widespread use of derived models.

Tree type ensemble algorithms were found to be successful in harmonizing different soil texture classification systems (Cisty et al., 2015), prediction of soil bulk density (Chen et al., 2018; Dharumarajan et al., 2017; Ramcharan et al., 2017; Sequeira et al., 2014; Souza et al., 2016), but have not been intensively applied yet to derive input parameters for hydrological modelling (Koestel and Jorda, 2014; Tóth et al., 2014).

Hengl et al. (2018a) tested several machine learning algorithms (i.e. neural networks, random forest, gradient boosting, K-nearest neighbourhood and cubist) to map potential natural vegetation. From those random forest performed the best. Nussbaum et al. (2018) analysed different methods to map several soil properties for three study sites in Switzerland. They also found that the random forest method performed the best when a single model was used. Adhikari et al. (2014) used the

cubist method combined with kriging for mapping soil organic carbon concentration and stock in Denmark and they found that cubist was appropriate for this purpose. The same was observed by Matos-Moreira et al. (2017), they used cubist for mapping the phosphorus concentration in north-western France. Behrens et al. (2018) compared a number of state of the art digital soil mapping methods including geostatistical techniques (i.e. ordinary kriging, regression kriging and geographically weighted regression), and machine learning algorithms (i.e. multivariate adaptive regression splines, radial basis function support vector machines, cubist, random forest and neural networks). They obtained the best results with cubist, random forest and bagged multivariate adaptive regression splines. Results of Rudiyanto et al. (2018) also showed that among several tested methods tree-based models performed the best. Hengl et al. (2018b) reviewed machine learning algorithms and geostatistical methods for soil mapping and found that the random forest method combined with the calculation of geographical proximity effects is a powerful method similarly to universal kriging.

Soil hydraulic maps are mostly derived by two ways i) by applying pedotransfer functions (PTFs) on available soil and/or environmental maps, called an indirect mapping method, ii) with direct spatial inference of observation point data (Bouma, 1989), which is considered to be a direct procedure. Point data can be measured or predicted by PTFs. Several studies analysed the efficiency of geostatistical methods to map water retention at specific matric potential (Farkas et al., 2008) and saturated hydraulic conductivity (Motaghian and Mohammadi, 2011; Xu et al., 2017). Ferrer Julià et al. (2004) mapped soil hydraulic conductivity for the Spanish area of the Iberian Peninsula at 1 km resolution with both methods (i) and (ii). They found that the map derived by kriging interpolation performed the best. Farkas et al. (2008) mapped water content at field capacity and wilting point with geostatsitical methods for an area of 1483 ha. They optimized sampling density needed to derive 10 m resolution soil hydraulic maps for their study site.

In most cases there is no available point data for applying geostatistical methods, therefore in several studies soil hydraulic maps were generated with a PTF applied on easily available spatial soil data (Chaney et al., 2016; Dai et al., 2013; Marthews et al., 2014; Montzka et al., 2017; Tóth et al., 2017; Wu et al., 2018).

Further to the spatial variability of soil hydraulic properties, information on the prediction uncertainty is important for modelling tasks. In this way extreme conditions might be better described. A possible calculation of this kind of uncertainty was provided by Montzka et al. (2017). They calculated sub-grid variability of the coupled Mualem-van Genuchten model parameters for a coarse 0.25° grid based on fitting water retention and hydraulic conductivity model for each grid cell of the 1 km resolution SoilGrids. Román Dobarco et al. (2019) and McNeill et al. (2018) also provided information on the uncertainty of the prediction of soil hydraulic properties. Root mean squared error (RMSE) of published PTFs predicting soil water retention is usually between 0.02 and 0.07 $cm^3$ $cm^{-3}$ depending on the predicted soil hydraulic property and available input information, e.g. in Nguyen et al. (2017), Zhang and Schaap (2017) or Román Dobarco et al. (2019) to mention some of the latest results. When PTFs are used for mapping, the uncertainty of the input soil layers will further increase the uncertainty of the calculated soil hydraulic properties, e.g. in point based validation RMSE was 0.073 $cm^3$ $cm^{-3}$ for water content at field capacity mapped for China in Wu et al. (2018); Leenaars et al. (2018) found that mean RMSE for water content at saturation,

field capacity and wilting point together was 0.102 cm$^3$ cm$^{-3}$ for African soils; in EU-SoilHydroGrids (Tóth et al., 2017) RMSE was 0.095, 0.096, 0.084 cm$^3$ cm$^{-3}$ for water content at saturation, field capacity and wilting point respectively for European soils.

Our aim was to analyse the performance of two different mapping methods in deriving 3D soil hydraulic properties, such as water content at saturation (THS), field capacity (FC) and wilting point (WP) on the Balaton catchment area in Hungary. Soil hydraulic maps were derived by i) an indirect method: applying local hydraulic PTFs on the available soil and other environmental spatial information of the catchment and ii) geostatistical – direct – method using available soil profile data and environmental covariates of the catchment. Performance of derived soil hydraulic maps was compared to that of the 3D
European soil hydraulic maps (EU-SoilHydroGrids v1.0) (Tóth et al., 2017).

## 2 Materials and methods

### 2.1 Study site

We selected the catchment area of Lake Balaton (Fig. 1) to study mapping of soil hydraulic properties, because it is an important area in Hungary from the point of modelling hydrological, ecological, meteorological processes or planning land
use and management. The size of the catchment is 5775 km$^2$. The mean depth of the lake is 3.5 m therefore water quality and quantity of the lake is sensitive to environmental changes. It has a warm temperate climate with 9-12°C mean annual temperature and 560-770 mm mean annual precipitation, lower temperature and higher rainfall values tend to be towards the western and elevated areas. Elevation is between 100 and 500 m on the northern part and 100 and 300 m in other areas of the catchment. Main soil types are Luvisols (53%), Cambisols (18%), Gleysols (10%), Histosols (5%) further to those Stagnosols,
Arenosols, Regosols, Leptosols and Chernozems also occur (IUSS Working Group WRB, 2014).

For the catchment spatial information on soil type, clay, silt and sand content, organic matter content, calcium carbonate content and pH in water (pH) at 100 m resolution was provided by the DOSoReMI.hu (Digital, Optimized Soil Related Maps and Information; (Pásztor et al., 2018b)) framework (Table 1). As soil chemical properties – organic matter content, calcium
carbonate content and pH – were only available for the 0-30 cm depth, those could only be considered for the topsoil predictions. Information on topography, meteorology, geology and vegetation listed in Table 1 was used as predictors and environmental covariates for the elaboration of PTFs and direct mapping accordingly.
Topographical parameters were calculated with SAGA GIS tools (Conrad et al., 2015) based on the digital elevation model. For the mapping of soil hydraulic properties all covariates were harmonized, projected to the Hungarian Uniform National
Projection system, rasterized if necessary and resampled to 100 m resolution.

## 2.2 Dataset to relate soil hydraulic properties and environmental information

For the prediction of soil hydraulic properties based on soil and other environmental variables the Hungarian Detailed Soil Hydrophysical Database (Makó et al., 2010) was used, extended with topographical, meteorological, geological information and remotely sensed vegetation properties (Table 1), called MARTHA ver 3.0 (acronym of the Hungarian name of the dataset). MARTHA consists of 15142 soil horizons' data belonging to 3970 soil profiles. The samples in it have measured information on basic soil properties – e.g. soil depth, organic matter content, clay, silt and sand content, calcium carbonate content, pH, etc. – and also on soil hydraulic properties such as soil water retention at different matric potential values.

## 2.3 Mapped soil hydraulic properties

We mapped soil water content at 0, -330 and -15,000 cm matric potential values, THS, FC and WP respectively, because these soil hydraulic properties are often required for various purposes. Definition of FC varies across different countries. In Hungary FC is determined at -330 cm matric potential, therefore water content at -100 or -200 cm was not analysed in the presented work.

The information on soil properties were available for 0-30, 30-60 and 60-90 cm soil depths and this determined the vertical resolution of the soil hydraulic maps. As PTFs include depth as independent variable, they are applicable for any soil depth intervals.

## 2.4 Methods for soil hydraulic properties mapping

Soil hydraulic properties were mapped both with direct and indirect methods for the catchment of Lake Balaton. In direct mapping the target soil variable is directly interpolated over the domain of interest, whereas in indirect mapping not the target variable but its components, factors, and/or covariates are interpolated first and then these interpolated surfaces are used to compute and map the target variable. In the direct method we used the geostatistical approach to spatially inference measured soil hydraulic data collected in profiles of the catchment through modelling its relationship with environmental covariates. In indirect mapping PTFs were derived first to describe relationships between soil hydraulic properties and easily available soil and other environmental parameters. In this approach the full national MARTHA database provided soil reference data, and nationwide, spatially exhaustive environmental auxiliary information was used. The PTF predictions were then spatially implemented on the environmental covariates clipped for the catchment area of Lake Balaton (Fig. 2).

### 2.4.1 Pedotransfer function based indirect mapping (HUN-PTF)

We derived PTFs for THS, FC and WP using soil depth, soil properties and other environmental covariates listed in Table 1 as independent variables. Organic matter content, calcium carbonate content and pH could be considered only for the topsoil (0-30 cm) predictions, because those are not available for the subsoils on the Balaton catchment area.

For the construction of PTFs those samples were selected from the MARTHA dataset which had measured values of soil horizons or layers considered as dependent and independent variables. We needed two kinds of predictions: (1) for topsoils where we could include Organic matter content, calcium carbonate content and pH among the predictors and (2) for subsoils without the above soil chemical parameters, because those are not available for the 30-60 and 60-90 cm soil depths on the Balaton catchment. First we randomly selected 67% of the samples from those which had data on the dependent and all the

independent variables available on the catchment area to derive the PTFs. The remaining 33% was used to compare the performance of the PTFs, this we called TEST_CHEM set. In the second step we needed a training set (67% of data) and a test set (33% of data) also for subsoil prediction for which we did not have to apply the restriction on the soil chemical properties, therefore we could include more samples for the analysis. As a test set we used the samples of the TEST_CHEM set and further added cases to reach the 33% of the complete data appropriate for subsoil predictions. Again the remaining 67% was used for

training.

The number of samples used to train and test the PTFs was 8,157 and 12039 for THS, 8,051 and 11,931 for FC, 8,195 and 12,036 for WP, with and without soil chemical properties respectively.

We analysed prediction performance of the two widely used machine learning algorithms, random forest (RF) of R package

'ranger' (Wright, Wager, & Probst, 2018) and generalized boosted regression model (GBM) of 'gbm' (Ridgeway, 2017) for the prediction of THS, FC and WP. The advantage of these two algorithms is that the prediction intervals of the dependent variable are computed as a function of the independent variables.

Both algorithms build ensembles of models from regression trees. In regression trees data is recursively partitioned to increase homogeneity in the subsets, in this way residual sum of squares are minimized (Breiman et al., 1984). The difference between

GBM and RF is the way the forest is built from the individual trees. RF relies on averaging the result of the trees in the ensemble. The trees are grown on $n_{tree}$ bootstrap samples of the training data independently from each other (Breiman, 2001), therefore it is a bagging type ensemble. At each split of the trees only a small set of predictors is selected randomly to analyse which variable at which split point is the best for the partition, i.e. minimize the sum of squares. In GBM the ensemble model is grown sequentially, at each iteration step the next model is built with respect to the error of the ensemble learnt so far

(Friedman, 2001; Natekin and Knoll, 2013), which is characteristic for the boosting type ensemble, already included in its name (Dietterich, 2000). In each split all possible predictors are considered.

Optimization of parameter set in RF and GBM model was performed with the train function of R package 'caret' (Kuhn et al., 2018). Five times repeated five-fold cross-validation was used to evaluate performance of different parameter sets. For RF

number of input parameters selected randomly at each split – which is set under the 'mtry' argument – was tuned. In the case of GBM influence of interaction depth and shrinkage were analysed. In ranger RF default value is 500 for the number of trees, that was used for both RF and GBM. Also for minimum number of observations in the terminal nodes of the trees the default value of the algorithms was used. During the tuning of model parameters the importance of variables was calculated both for GBM and RF methods to eliminate the less relevant predictors (Gregorutti et al., 2017; Nussbaum et al., 2018). Variable importance is the measure of relevance of each predictor, it is calculated from the average sum of squared improvements at each split, where the predictor was selected to partition the data (Hastie et al., 2009). A value of 100 is assigned to the largest variable importance value and the others are scaled accordingly to provide relative measure. The most important 50-50 predictors out of 173 for topsoils and 170 for subsoils have been selected from both GBM and RF models. After concatenating the 50-50 most important variables parameter tuning was performed again with the decreased number of predictors. We compared the accuracy of all models based on the cross-validation results and built the final prediction model (PTF) with the better performing and simpler algorithm on all training data with the optimized parameters. Performance of the PTFs was RMSE (Eq. 1) and coefficient of determination ($R^2$) (Eq. 2).

$$RMSE = \sqrt{\frac{1}{N}\sum_{i=1}^{N}(y_i - \hat{y}_i)^2} = \sqrt{MSE} \tag{1}$$

$$R^2 = 1 - \frac{\sum_{i=1}^{N}(y_i - \hat{y}_i)^2}{\sum_{i=1}^{N}(y_i - \bar{y})^2} \tag{2}$$

Performance of PTFs on the training dataset was based on the results of a five-fold cross-validation, and out-of-bag samples – not included in the bootstrap sample used to build the tree – for GBM and RF respectively. In RF accuracy on out-of-bag samples was analysed. Uncertainty of the predictions was characterized with the 5 and 95% quantiles of the predicted values, calculated within the 'ranger' and 'gbm' packages during the derivation of the prediction algorithms.

HUN-PTFs derived on the MARTHA dataset were used to calculate the soil hydraulic properties (THS, FC, WP) based on the available soil and environmental covariates available for the catchment (Table 1, section 2.1) as predictors, hence those were mapped indirectly. Soil information is currently available for the 0-30, 30-60 and 60-90 cm. The input information depth was set to 15, 45 and 75 cm for the first, second and third layer respectively during the calculation of soil hydraulic property maps.

We provided information on the uncertainty of the predictions by pixels. Further to the median and the 5 and 95% quantiles of the predicted values were also mapped for each soil hydraulic property. The prediction intervals were calculated by the PTFs.

## 2.4.2 Direct mapping with geostatistical method (RFK)

We applied random forest combined with kriging (RFK), which can be considered as a new 'workhorse' of digital soil mapping (Keskin and Grunwald, 2018). In the case of RFK, the deterministic component of spatial soil variation is modelled by the RF introduced above, whereas the stochastic part of variation is modelled by kriging using the derived residuals.

For the geostatistical analysis those samples of the MARTHA database were selected which fall within the catchment plus a 5 km buffer zone area. The buffer zone was used to increase the accuracy of geostatistical calculations also at the border of the catchment. On the study site data of 359 soil profiles are available from the MARTHA (Fig. 3). Table 2 summarizes the measured soil chemical, physical, hydraulic data of the soil profiles' horizons.

First of all, we harmonized the soil hydraulic dataset for the required soil depths (i.e. 0-30, 30-60, 60-90 cm) by using equal-area splines (Malone et al., 2009), then we used RFK for predicting each soil hydraulic property for each soil depth, respectively. For RF we also optimized the parameter set by the 'train' function of R package 'caret' using five times repeated five-fold cross-validation. The most important 50 covariates – out of 173 for topsoils and 170 for subsoils, listed in Table 1 – have been selected and the final RF model was optimized with those predictors. We used the final RF model for predicting the

deterministic component. We computed the residuals and then we estimated their variogram by Matheron's (1963) method-of-moments estimator. An isotropic variogram model was fitted to the estimated variogram by the 'fit.variogram' function of R package 'gstat' (Gräler et al., 2016; Pebesma, 2004). We kriged the residuals and then we added them to the deterministic component predicted by RF. The above described modelling procedure was applied for each soil hydraulic property and for each soil depth. Performance of RF was described with RMSE (Eq. 1) and $R^2$ (Eq. 2).

## 2.4.3 Evaluating the performance of soil hydraulic maps

Performance of soil hydraulic maps was evaluated based on observed soil hydraulic properties harmonized for 0-30, 30-60 and 60-90 cm depth with the method described in 2.4.2 section. RMSE and mean square error skill score ($SS_{mse}$) (Nussbaum et al., 2018) Eq. (1-3) were calculated for each map.

$$SS_{mse} = 1 - \frac{\sum_{i=1}^{N}(y_i - \hat{y}_i)^2}{\sum_{i=1}^{N}\left(y_i - \frac{1}{N}\sum_{i=1}^{N} y_i\right)^2} \tag{3}$$

Performance of soil hydraulic maps derived with HUN-PTFs and RFK was compared to the 3D European soil hydraulic maps (EU-SoilHydroGrids v1.0) (Tóth et al., 2017). In EU-SoilHydroGrids input information for mapping was SoilGrids 250 m (Hengl et al., 2017) on which EU-PTFs (Tóth et al., 2015) were applied, hence its resolution is 250 m. We converted the information of EU-SoilHydroGrids to 0-30, 30-60 and 60-90 cm to be able to compare its performance to the 100 m resolution new soil hydraulic maps derived by HUN-PTFs and RFK.

The Kruskal Wallis test implemented in the R package 'agricolae' (De Mendiburu, 2017) was applied at 5% significance level on the mean squared error values for the comparison of the PTFs with different input variables and also the soil hydraulic maps derived using different methods.

All statistical analyses were performed in R (R Core Team, 2017).

## 3 Results and discussion

### 3.1 Pedotransfer functions

During the parameter tuning of RF and GBM we found that decreasing number of input variables – from 173 to 69-76 and from 170 to 65-77 in case of topsoil and subsoil predictions respectively – significantly improved prediction of top- and subsoil

FC and subsoil WP. Although differences between RMSE values were less than 0.0001 $cm^3$ $cm^{-3}$, these are negligible from a practical point of view. In Nussbaum et al. (2018) the number of input parameters were decreased from 300-500 environmental covariates to the 10, 20, 30, 40, 50 most important ones. No changes in performance were found during validation. We can assume that performance of predictions will neither increase nor decrease if more important independent variables are used exclusively for the predictions. Although the selection of the most important independent variables can reduce (i) unnecessarily

large size of the model which can speed up mapping of soil hydraulic properties for larger areas at fine resolution and (ii) multicollinearity between predictor variables. Dorman et al. (2013) extensively studied the problem of collinearity to test its impact on predictions of ecological parameters. They analysed multiple regression and machine-learning methods and found that prediction performance of random forest did not get worse due to high collinearity in the training dataset even when structure of collinearity was different in training and validation data. Influence of multicollinearity on the prediction

performance is partly reduced due to the random selection features of RF but could be further elaborated in the presented methods, however this was beyond the scope of the presented work.

In the case of RF optimal number of input parameters randomly selected at each split was between 10 and 20, depending on soil hydraulic parameter. In GBM optimal interaction depth varied between 20 and 40. Iteration converged during the prediction of lower 5% and upper 95% quantiles, but did not for 50%, which is the most probable predicted value. Therefore,

the influence of shrinkage and increasing the number of trees to 1,000 was also analysed but only in the prediction of FC because training with low shrinkage values is very time consuming. We tuned shrinkage 0.1 and 0.01 with both 500 and 1,000 trees, setting interaction depth to 4, 6 and 10. Shrinkage with 0.1 value was more accurate than 0.01 independently from the number of trees and increasing number of trees did not significantly improve the prediction, therefore shrinkage was set to 0.1 and the default 500 number of trees were used in the algorithm.

Performance of PTFs derived by RF and GBM on training and test sets is included in Table 3. In the case of all soil hydraulic properties RF performed significantly better than GBM based on MSE on TEST and TEST_CHEM sets both for topsoil and subsoil predictions, except for WP topsoil predictions, where there was no significant difference between the methods. In this way PTFs derived with RF method were selected for mapping soil hydraulic properties. RMSE values calculated on the test sets for RF were between 0.042 and 0.045 $cm^3$ $cm^{-3}$ for THS, 0.039 and 0.042 $cm^3$ $cm^{-3}$ for FC, 0.035 and 0.038 $cm^3$ $cm^{-3}$ for WP, which is close to the performance of other internationally accepted PTFs (e.g. Botula et al. (2013), Román Dobarco et al. (2019), Zhang and Schaap (2017)). $R^2$ was 0.408-0.487, 0.746-0.766 and 0.737-0.762 for THS, FC and WP respectively on test sets in the case of RF. Figure 4. shows the scatterplots of measured versus predicted values with the 90% prediction interval. At the lower end of the soil hydraulic property distribution, real values were closer to the lower 5% quantile predictions, at the higher end of its distribution the real values are closer to the upper 95% quantile predictions. When we compared performance of RF derived for topsoils – which includes organic matter content, pH and calcium carbonate content as well among the input parameters - and subsoils there was no significant difference based on the results in the TEST_CHEM set. This is due to their correlation with other environmental predictors considered in the PTFs such as soil texture, depth, longitude, elevation, slope angle, multi-resolution valley bottom flatness, horizontal distance to existing water bodies, roughness, temperature, precipitation, solar radiance, spectral reflectance in red and near infrared and normalized difference vegetation index (Adhikari et al., 2014; Hengl et al., 2017; Nussbaum et al., 2018). When other environmental covariates than soil related variables are not included among input parameters chemical properties significantly improve prediction (Hodnett and Tomasella, 2002; Khodaverdiloo et al., 2011; Tóth et al., 2015). In the case of THS range, the predicted values using chemical parameters as well were closer to the range of measured values, therefore we also considered soil chemical properties for the topsoil predictions. For FC and WP range of values predicted with PTF not including chemical variables were closer to that of measured values, hence information on organic matter content, pH and calcium carbonate content – even though it is available – was not considered during the estimation of topsoil hydraulic properties.

The presented PTFs were derived on the full MARTHA dataset, therefore those are applicable to predict the THS, FC and WP of soils in the whole Pannonian region.

### 3.1.1 Importance of independent variables

For THS organic matter content, silt, sand content, pH, clay, calcium carbonate content are the most important variables with relative importance of over 20% based on final RF model. Further to those properties, soil depth, mean annual precipitation, mean monthly maximum, minimum and mean temperature of some months, mean monthly radiation, longitude, horizontal and vertical distance to existing water bodies, multi-resolution valley bottom flatness and ridge top flatness, water vapour pressure in August, spectral reflectance in near infrared are among the most important 30 variables having 10-15 % relative importance. For FC and WP clay, silt and sand content and organic matter content are the most important variables, having

relative importance around and over 20 %. Soil type, mean monthly precipitation in July, vertical distance to existing water bodies and longitude have relative importance around 5-14 % in case of FC. All the other environmental covariates have relative importance of less than 5%. For WP longitude, mean monthly precipitation of November and July, elevation, vertical and horizontal distance to existing water bodies, calcium carbonate content, mean monthly radiation, pH, depth, mean monthly water vapour pressure, multi-resolution ridge top flatness and spectral reflectance in near infrared have relative importance of between 5-16 %. Information on topography was found important for the prediction of soil hydraulic properties by Obi et al. (2014), Rawls and Pachepsky (2002), Romano and Chirico (2004), Zhao et al. (2016) as well. Information on land cover was not retained after selecting the most important variables.

When soil chemical properties (organic matter content, calcium carbonate content, pH) are not included among input parameters, sand, silt, clay content are far the most important three independent variables (39-100 %). In the case of THS also depth has higher relative importance (52 %). For the prediction of FC importance of soil type increases to 18 %. In WP prediction there is no notable change in variable importance when chemical properties are not included in the RF.

Summary of the variable importance analysis showed that soil properties are far the most important input parameters for the prediction of soil hydraulic properties (Fig. 5). In this way resolution of soil maps determined the resolution of the derived soil hydraulic maps, which was 100 m.

## 3.2 Random forest combined with kriging (RFK)

During the RF parameter tuning we also found that decreasing the number of environmental covariates – from 173 to 50 and from 170 to 50 in the case of topsoil and subsoil respectively – significantly improved the prediction accuracy for each soil hydraulic property. For the final RF models the optimal number of randomly selected predictors at each split varied between 5 and 40 depending on the given soil hydraulic property. The performance of the final RF models are summarized in Table 4. $R^2$ varies between 0.189-0.403, 0.478-0.562 and 0.463-474 for THS, FC and WP, respectively. RMSE was 0.055-0.060, 0.053-0.063 and 0.051-0.056 for THS, FC and WP, respectively. For describing spatial variation of the soil hydraulic properties the most important environmental covariates were the soil type, organic matter content (for topsoil), clay, silt and sand content and the pH (for topsoil). The final RF models were used for estimating the deterministic component for each soil hydraulic property.

The parameters of the fitted variogram models are summarized in Table 4. In the case of exploratory variography most of the experimental variograms did not show spatial structure and the applied variogram fitting algorithm was not able to find a satisfactory variogram model in case of six out of nine under 200 iterations. Hence, a nugget model was fitted to those variograms (Table 4), which is not rare in digital soil mapping (Hengl et al., 2015; Szatmári and Pásztor, 2018; Vaysse and Lagacherie, 2017). In Table 4 we have observed that the lower the $R^2$ value was, the higher the range parameter became. The fitted variogram models were used for kriging of the RF residuals for each soil hydraulic property. We summed the RF predictions and the kriged residuals to get the RFK maps for each of the target hydraulic properties.

## 3.3 Performance of soil hydraulic maps

New 100 m resolution soil hydraulic maps significantly outperformed the EU-SoilHydroGrids (Table 5), which was expected because (i) reference soil data originate from the mapped area, (ii) also spatially denser and (iii) locally trained models are used. In addition, several environmental covariates were considered for the predictions and relationship between easily available soil properties and soil hydraulic parameters were derived from local data.

In the case of mapping six out of nine soil hydraulic maps there was no significant difference between maps derived by RFK and HUN-PTFs. In the case of THS HUN-PTF performed significantly better for mapping the 30-60 and 60-90 cm. For calculating WP at 60-90 cm soil depth RFK was significantly better than HUN-PTF method.

The range of predicted values is smaller in the case of HUN-PTF method than in RFK, which is due to the "averaging approach" of the algorithm which in the case of RFK is spatially corrected allowing a wider range in the predicted values (Fig. 6, 7, 8, 9). Density plot of predicted values are smoother in the case of RFK than in HUN-PTF and EU-SoilHydroGrids maps (Fig. 6). This is due to adding residuals of kriging which modifies the values derived by random forest. In EU-SoilHydroGrids soil hydraulic values were calculated with linear regression based on soil properties available from SoilGrids, where mapping was performed with RF without kriging. In this way possible soil input combinations are limited in the European maps. In SoilGrids algorithms are derived from a global dataset (Hengl et al., 2017), which has sparser measured data than the Hungarian soil profile database used to map soil properties (Laborczi et al., 2018; Szatmári and Pásztor, 2018). In addition, RF is based on an averaging algorithm, which limits the ability to describe local extreme values. These result in smaller range and variability of calculated soil hydraulic properties on EU- SoilHydroGrids maps than on RFK or HUN-PTF ones (Fig. 6) The basic Hungarian soil maps were derived with regression kriging methods, thus providing smoother soil input data for the calculations. As an example of how differences in the range of predicted soil hydraulic properties can be visualized, the maps of THS, FC, WP are shown on Fig. 7-9 (a), (b), (c) for a selected area of the catchment. Differences between the new and already available maps also comes occur due to the differences in resolution, which is 100 m for RFK and HUN-PTF and 250 m for EU-SoilHydroGrids. Even though the influence of topographical information was less than that of soil properties when PTFs were derived, the pattern of topography is visible on the maps derived by RFK and HUN-PTFs. This is due to the soil layers used as inputs for calculating the soil hydraulic properties, because topographical information was important among the covariates when the maps on them were derived (Szatmári et al., 2013). In RFK influence of the topography is less visible, it could be smoothed by adding kriged residuals. A map of possible lower 5 % and upper 95 % values based on the HUN-PTF method are also shown in Fig. 7-9 (d), (e). The range between the lower and upper possible values (Fig. 10) are usually higher for Histosols, Gleysols and Luvisols under forest land use, because these kind of soils are underrepresented in the MARTHA database.

Although we compared the performance of the new soil hydraulic maps to that of EU-SoilHydroGrids, it was out of our aims to differentiate the uncertainty of the maps originating from the soil input layers – i.e. DOSoReMI.hu and SoilGrids.

Average difference between the RFK and HUN-PTFs maps is between 0.003 and 0.012 $cm^3$ $cm^{-3}$ for THS, 0.011 and 0.015 $cm^3$ $cm^{-3}$ for FC, 0.015 and 0.018 $cm^3$ $cm^{-3}$ for WP, depending on soil depth. Absolute difference between the maps derived with HUN-PTFs and RFK is less than 0.025 $cm^3$ $cm^{-3}$ for at least 65 % of the mapped area and was always smaller than 0.100 $cm^3$ $cm^{-3}$ (Table 6). On those areas where difference between RFK and HUN-PTF was higher than 0.025 $cm^3$ $cm^{-3}$, HUN-PTF predicted lower water retention at all matric potential values for Histosols and Luvisols under forest land use type. WP values predicted with HUN-PTFs were higher than that of RFK for Luvisols with sandy texture and under forest land use type.

Based on $SS_{mse}$ values in the case of seven out of nine soil hydraulic maps RFK mapping method was more accurate than HUN-PTF, although only calculation of WP in 60-90 cm depth was significantly better. For THS HUN-PTFs performed significantly better at 30-60 and 60-90 cm soil depth.

In this study priority was put on the usability and transferability of the results into practical applications. The purpose of the presented research was to derive as accurate maps as possible. Thus ability for full comparability of the methods did not determine design of methodology and statistical analysis. Therefore, in the RFK analysis all measured data were used for the mapping. For the PTF approach predictions were tested on randomly selected 33% samples of the whole MARTHA database without distinguishing samples located on the catchment, as it is usually done in deriving PTFs. This provides broader information and possibility for a wider application of PTFs. The presented HUN-PTF mapping method can be applied in any catchments of Hungary.

### 3.4 Practical use of the analysis

RF performed significantly better than GBM in 7 cases out of 8 on test sets. RF was found to be a suitable method to provide information on the prediction uncertainty, any desired quantiles of the predicted value can be calculated. This enables it to include extreme soil hydraulic parameters for hydrological simulations. Its further advantage is that it can handle several independent variables, performance of prediction is not influenced by multicollinearity between independent variables and inclusion of unimportant input parameter. Calculation on multiple cores is implemented in the random forest algorithm in 'ranger' R package, which can significantly decrease computation time.

Easily available soil properties such as sand, silt and clay content, organic matter content and depth were the most important input variables for the calculation of THS, FC and WP among the analysed 173 soil and environmental covariates. For THS calcium carbonate content and pH were also among independent variables with higher importance. Geographical coordinates, information on topography, climate and vegetation had smaller relative importance. Covariates on land use and parent material were not among the 50 most important variables. Therefore, resolution of available soil maps determined the resolution of new soil hydraulic maps, which is 100 m.

The number of input variables can be decreased based on variable importance, which can significantly decrease computation time and information not relevant for the prediction can be discarded. For practical application it is desirable to decrease the size of the prediction models when PTFs are applied for soil hydraulic mapping at country scale at finer resolution.

If data on topography, climate and vegetation are also considered for the prediction missing information on chemical properties, such as organic matter content, pH, calcium carbonate content can be covered by the environmental covariates without significant loss of performance.

HUN-PTFs performed significantly better for the prediction of THS at 30-60 and 60-90 cm depth, although the absolute difference between the RFK and HUN-PTFs maps is less than 0.025 $cm^3$ $cm^{-3}$ for at least 75 % of the area. Spatial patterns of topography are less dominant on the soil hydraulic maps prepared by the RFK method due to kriging the residuals, which is an advantage. Maps prepared by the HUN-PTFs cannot decrease the influence of topography included in the input layers therefore even if topographical parameters are not important for the prediction of soil hydraulic properties that are visible on the soil hydraulic maps. Considering all these results we suggest using the soil hydraulic maps prepared by the RFK only if the most probable soil hydraulic value is needed for the Balaton catchment area. Information on the uncertainty of the predicted values can be derived with geostatistical methods as well, e.g. Szatmári and Pásztor (2018), Rudiyanto et al. (2016), Viscarra Rossel et al. (2015) presented possible methods. According to Szatmári and Pásztor (2018), quantile regression forest (Meinshausen, 2006) based uncertainty quantification outperforms most of the prediction techniques used in digital soil mapping. Furthermore, they have pointed out that bootstrapping based uncertainty quantification for RFK is quite time consuming, as well as requiring massive storage and computing capacity. The ranger package - with which we derived the HUN-PTFs - includes implementation of quantile regression forest (Meinshausen, 2006) for the calculations of the prediction intervals. If information on uncertainty is needed as well maps derived by the HUN-PTFs are recommended to use. In Table 7 we highlighted the most important differences between pedotransfer function (HUN-PTF) and geostatistical (RFK) based soil hydraulic mapping based on the Balaton catchment. Most of the findings are in line with Hengl et al. (2018b), Tranter et al. (2009), Vaysse and Lagacherie (2017), Webster and Oliver (2007).

**4 Conclusions**

Based on results of six out of nine soil hydraulic maps there is no significant difference in performance between pedotransfer function (indirect) and geostatistical (direct) method on the Balaton catchment area. The benefit of maps computed with random forest and kriging is that locally extreme values can be characterized better. In the case of pedotransfer function based mapping it is advantageous that calculation of uncertainty is much less computation intensive than it is with geostatistical methods, although it would be interesting in the future to analyse the difference between uncertainty maps calculated with the different methods specifically for soil hydraulic properties.

**Data availability.** The 3D soil hydraulic maps of the Balaton catchment – in GeoTIFF format – and the hydraulic pedotransfer functions – in RData format – are freely available for non-commercial use from the Institute for Soil Sciences and Agricultural Chemistry Centre for Agricultural Research Hungarian Academy of Sciences (http://mta-taki.hu/en/kh124765/maps, https://www.mta-taki.hu/en/kh124765/hun_ptfs).

**Author contribution.** BT conceptualized the study, designed the methodology and coordinated the research. AM provided the MARTHA dataset. LP and AL cured soil maps, KT prepared all the other covariate layers. AM, LP, KT, AL, GSZ, BT performed data curation. GSZ carried out geostatistical analysis, BT derived the PTFs, they applied statistical and computational analysis. AL assisted in visualization of maps and built website for data download. KR, AM, LP contributed to the interpretation. LP provided the computing resources. BT prepared the paper with considerable input from GSZ and further
contributions from all co-authors.

**Competing interests.** The authors declare that they have no conflict of interest.

**Acknowledgements.** The research project was supported by the Hungarian National Research, Development and Innovation Office (NRDI) under grants KH124765, KH126725, K119475, through the common grant of the Hungarian and Polish Academy of Sciences (Grant No. NKM-108/2017)" and the János Bolyai Research Scholarship of the Hungarian Academy of
Sciences.

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

Table 1. Available environmental covariates.

| Name | Resolution | Description |
| --- | --- | --- |
| **Soil** | | |
| soil type | 100 m | according to Hungarian classification system (Pásztor et al., 2018a) |
| clay, silt, sand content | 100 m | 0-30, 30-60, 60-90 cm (Laborczi et al., 2018) |
| organic matter content | 100 m | 0-30 cm (Szatmári and Pásztor, 2018) |
| calcium carbonate content | 100 m | 0-30 cm (Pásztor et al., 2018b) |
| pH in water | 100 m | 0-30 cm (Pásztor et al., 2017) |
| **Parent material** | 1:100000 | (Gyalog and Síkhegyi, 2005), map was converted to raster layer |
| **Topography** | | |
| digital elevation model | 25 m | (Bashfield and Keim, 2011) |
| | | elevation, slope angle, aspect, northing and easting aspects, planar curvatures, profile curvatures, combined curvatures, topographic position indices, topographic position indices, terrain ruggedness indices, roughness, dissection, surface to area ratio, multi-resolution valley bottom flatness, multi-resolution ridge top flatness, negative openness, positive openness, convergence indices, LS factor, vector ruggedness measure, surface convexity, flow accumulation area, flow length, topographic wetness indices by single and multi-flow algorithms, vertical distance to existing water bodies, vertical distance to existing water bodies, horizontal distance to existing water bodies, smoothed version of elevation, smoothed version of profile curvature, smoothed version of slope, smoothed version of total curvature, standard deviations of elevation, standard deviations of profile curvature, standard deviations of slope, standard deviations of total curvature |
| **Climate** | | |
| WorldClim | 30" | (Fick and Hijmans, 2017) |
| | | mean monthly temperature, precipitation, solar radiation, water vapour pressure, mean monthly minimum and maximum temperature |
| Hungarian data | 100 m | (Szentimrey and Bihari, 2007) |
| | | The spatial layers were compiled using the MISH method elaborated for the spatial interpolation of surface meteorological elements based on a 30 year observation by the Hungarian Meteorological Service with 0.5' resolution. mean annual precipitation and temperature |
| **State of vegetation** | | |
| MODIS | 250 m | (Vermote, 2015) normalized difference vegetation index, near infrared, red |
| **Land cover** | | |
| Copernicus Pan-European High Resolution Layers | 20 m | (CEC EEA, 2012) tree cover density, forest type, impermeable cover of soil, wetland, grassland |
| CORINE Land Cover | 25 ha | (CEC EEA, 2012) natural grassland, land principally occupied by agriculture |

Table 2. Description statistics of measured soil properties of the Balaton catchment

| Soil propety | N | Minimum | Maximum | Mean | SD | Median |
|---|---|---|---|---|---|---|
| Clay content (100 g g$^{-1}$) | 1453 | 0.00 | 79.43 | 21.27 | 9.38 | 20.29 |
| Silt content (100 g g$^{-1}$) | 1349 | 0.36 | 73.99 | 38.48 | 16.11 | 40.92 |
| Sand content (100 g g$^{-1}$) | 1349 | 2.85 | 95.94 | 40.37 | 21.48 | 35.09 |
| Organic matter content (100 g g$^{-1}$) | 1269 | 0.00 | 28.93 | 1.18 | 1.57 | 0.73 |
| Calcium carbonate content (100 g g$^{-1}$) | 925 | 0.00 | 72.00 | 9.75 | 11.97 | 4.50 |
| pH in water (-) | 1445 | 3.61 | 9.38 | 7.14 | 0.98 | 7.29 |
| Saturated water content (cm$^3$ cm$^{-3}$) | 1299 | 0.324 | 0.883 | 0.469 | 0.066 | 0.461 |
| Water content at field capacity (cm$^3$ cm$^{-3}$) | 1294 | 0.032 | 0.640 | 0.314 | 0.083 | 0.320 |
| Water content at wilting point (cm$^3$ cm$^{-3}$) | 1284 | 0.006 | 0.462 | 0.167 | 0.075 | 0.160 |

Table 3. Performance of hydraulic PTFs on training and test datasets. THS: saturated water content, FC: field capacity, WP: wilting point, RF: random forest method, GBM: generalized boosted regression method, TEST_CHEM set: test dataset in which chemical soil properties are available for the predictions, TEST set: test dataset, in which chemical soil properties are not necessarily available for the predictions, RMSE: root mean square error, $R^2$: determination coefficient.

| Predicted soil hydraulic property | Selected method* | Train set** | | | TEST set | | | TEST_CHEM set | | |
|---|---|---|---|---|---|---|---|---|---|---|
| | | $R^2$ | RMSE (cm³ cm⁻³) | N | $R^2$ | RMSE (cm³ cm⁻³) | N | $R^2$ | RMSE (cm³ cm⁻³) | N |
| **THS** topsoil | GBM | 0.453 | 0.052 | 5709 | - | - | - | 0.484 | 0.042 | 2448 |
| | RF | 0.488 | 0.041 | 5709 | - | - | - | 0.487 | 0.042 | 2448 |
| subsoil | GBM | 0.429 | 0.045 | 8428 | 0.418 | 0.045 | 3611 | 0.400 | 0.046 | 2448 |
| | RF | 0.480 | 0.043 | 8428 | 0.429 | 0.045 | 3611 | 0.408 | 0.045 | 2448 |
| **FC** topsoil | GBM | 0.714 | 0.043 | 5635 | - | - | - | 0.770 | 0.039 | 2416 |
| | RF | 0.736 | 0.041 | 5635 | - | - | - | 0.766 | 0.039 | 2416 |
| subsoil | GBM | 0.738 | 0.044 | 8352 | 0.739 | 0.042 | 3579 | 0.751 | 0.040 | 2416 |
| | RF | 0.756 | 0.042 | 8352 | 0.746 | 0.042 | 3579 | 0.759 | 0.040 | 2416 |
| **WP** topsoil | GBM | 0.722 | 0.038 | 5736 | - | - | - | 0.739 | 0.037 | 2459 |
| | RF | 0.736 | 0.037 | 5736 | - | - | - | 0.762 | 0.035 | 2459 |
| subsoil | GBM | 0.717 | 0.041 | 8425 | 0.716 | 0.039 | 3611 | 0.711 | 0.038 | 2459 |
| | RF | 0.747 | 0.039 | 8425 | 0.737 | 0.038 | 3611 | 0.744 | 0.036 | 2459 |

5  * Input parameters included in all analysis for topsoils: soil type according to Hungarian classification system, sand (50–2000 μm), silt (2–50 μm) and clay content (<2 μm) (100 g g⁻¹), mean depth (cm) and information on topography, vegetation, meteorology and parent material listed in Table 1. For subsoils organic matter content (100 g g⁻¹); pH in water and calcium carbonate content (100 g g⁻¹) were included as well.
** Prediction error calculated on training is based on out of bag error in case of RF and 5-fold cross-validation in case of GBM method.

Table 4. Performance of random forest method and parameters of the fitted variogram models during the geostatistical mapping approach.

| Predicted soil hydraulic properties | Depth (cm) | Random forest | | | Variogram | | | |
|---|---|---|---|---|---|---|---|---|
| | | $R^2$ | RMSE ($cm^3$ $cm^{-3}$) | N | Partial sill | Type | Range | Nugget |
| **THS** | 0-30 | 0.403 | 0.055 | 324 | 0 | „Nug" | - | 32.552 |
| | 30-60 | 0.251 | 0.055 | 321 | 11.037 | „Exp" | 1531 | 18.357 |
| | 60-90 | 0.189 | 0.060 | 315 | 14.150 | „Exp" | 8211 | 27.067 |
| **FC** | 0-30 | 0.562 | 0.053 | 324 | 0 | „Nug" | - | 29.895 |
| | 30-60 | 0.532 | 0.056 | 321 | 0 | „Nug" | - | 26.539 |
| | 60-90 | 0.478 | 0.063 | 315 | 0 | „Nug" | - | 32.356 |
| **WP** | 0-30 | 0.463 | 0.052 | 324 | 0 | „Nug" | - | 23.689 |
| | 30-60 | 0.474 | 0.051 | 321 | 0 | „Nug" | - | 22.655 |
| | 60-90 | 0.466 | 0.056 | 315 | 32.718 | „Sph" | 2149 | 0 |

Table 5. Performance of soil hydraulic maps derived by random forest and kriging method (RFK), Hungarian pedotransfer functions (HUN-PTF) and from EU-SoilHydroGrids 250m dataset (EU-SHG) on the Balaton catchment. RMSE: root mean square error, $SS_{mse}$: mean square error skill score.

| Predicted soil hydraulic property | Depth | Method | N | RMSE $(cm^3\ cm^{-3})$ | $SS_{mse}$ | Sign. difference* |
|---|---|---|---|---|---|---|
| **THS** | 0-30 cm | RFK | 324 | 0.056 | 0.382 | b |
| | | HUN-PTF | 350 | 0.067 | 0.118 | b |
| | | EU-SHG | 348 | 0.070 | 0.041 | a |
| | 30-60 cm | RFK | 321 | 0.060 | 0.119 | a |
| | | HUN-PTF | 345 | 0.058 | 0.150 | b |
| | | EU-SHG | 343 | 0.063 | -0.004 | a |
| | 60-90 cm | RFK | 315 | 0.063 | 0.112 | b |
| | | HUN-PTF | 337 | 0.060 | 0.171 | c |
| | | EU-SHG | 335 | 0.071 | -0.149 | a |
| **FC** | 0-30 cm | RFK | 324 | 0.053 | 0.547 | b |
| | | HUN-PTF | 350 | 0.067 | 0.265 | b |
| | | EU-SHG | 348 | 0.076 | 0.070 | a |
| | 30-60 cm | RFK | 321 | 0.057 | 0.515 | b |
| | | HUN-PTF | 345 | 0.069 | 0.278 | b |
| | | EU-SHG | 343 | 0.084 | -0.069 | a |
| | 60-90 cm | RFK | 315 | 0.062 | 0.485 | b |
| | | HUN-PTF | 337 | 0.074 | 0.232 | b |
| | | EU-SHG | 335 | 0.095 | -0.243 | a |
| **WP** | 0-30 cm | RFK | 324 | 0.052 | 0.453 | b |
| | | HUN-PTF | 349 | 0.062 | 0.244 | ab |
| | | EU-SHG | 347 | 0.071 | -0.038 | a |
| | 30-60 cm | RFK | 321 | 0.052 | 0.467 | b |
| | | HUN-PTF | 344 | 0.065 | 0.152 | b |
| | | EU-SHG | 342 | 0.074 | -0.112 | a |
| | 60-90 cm | RFK | 315 | 0.057 | 0.443 | c |
| | | HUN-PTF | 335 | 0.067 | 0.208 | b |
| | | EU-SHG | 333 | 0.076 | -0.026 | a |

*Different letters indicate significant differences at 0.05 level between the accuracy of the methods based on squared error, e.g. performance indicated with letter c is significantly better than the one noted with letter b and a.

Table 6. Proportion of mapped area having smaller than 0.025, 0.025-0.050, 0.05-0.100 and bigger than 0.10 cm$^3$ cm$^{-3}$ absolute difference between predicted soil hydraulic values derived by geostatistical method (RFK) and applying pedotransfer functions on local soil and environmental covariates (HUN-PTF).

| Absolute difference between RFK and HUN-PTF (cm$^3$ cm$^{-3}$) | Depth (cm) | % of mapped area | | |
|---|---|---|---|---|
| | | THS | FC | WP |
| 0-0.025 | 0-30 | 76 | 80 | 71 |
| | 30-60 | 86 | 77 | 65 |
| | 60-90 | 75 | 72 | 71 |
| 0.025-0.050 | 0-30 | 21 | 17 | 25 |
| | 30-60 | 10 | 21 | 26 |
| | 60-90 | 21 | 22 | 24 |
| 0.050-0.100 | 0-30 | 3 | 3 | 4 |
| | 30-60 | 4 | 2 | 9 |
| | 60-90 | 4 | 6 | 5 |
| 0.100 < | 0-30 | 0 | 0 | 0 |
| | 30-60 | 0 | 0 | 0 |
| | 60-90 | 0 | 0 | 0 |

Table 7. Differences between pedotransfer function based (PTF) and geostatistical (RFK) mapping methods based on calculating saturated water content, field capacity and wilting point for the Balaton catchment.

| Aspects of mapping | Differences between the soil hydraulic mapping methods | |
| --- | --- | --- |
| | **PTF – indirect method** | **RFK – direct method** |
| Main steps of mapping | 1. derive PTFs on available soil hydraulic dataset or use an appropriate PTF available from the literature, 2. apply PTFs on available environmental covariates | 1. harmonize soil profile dataset available for the mapping based on required soil depth, 2. predict deterministic component, 3. calculate the residuals, estimate their variograms, krige them, 4. add kriged residuals to the deterministic component |
| Dataset used to describe relationship between soil hydraulic data and covariates | - any soil hydraulic dataset which is hydropedologically similar to the area for which soil hydraulic maps are required<br><br>- advantages: mapping can be applied even if no soil hydraulic data is available for the study area; available PTF can also be used<br><br>- disadvantages: a soil hydraulic dataset is needed which has to be similar to data of the study site from soil hydropedological point of view; or if PTF is already available the soil hydrological dataset used to train the PTF has to be similar to the study site | - soil hydraulic data available for the catchment<br><br>- advantages: soil hydraulic data is characteristic for the study site, locally extreme values can be better characterized<br><br>- disadvantages: density of measured soil hydraulic properties available for the study site might not satisfy the needs for mapping; further to the soil property, which is mapped, measured data of soil properties used in the prediction of the deterministic component (e.g. particle size distribution, organic matter content) is required as well |
| Inclusion of soil depth | - can be included as independent variable<br><br>- advantages: measured soil hydraulic properties are related to measured soil properties; soil hydraulic properties at any depth can be calculated<br><br>- disadvantages: certain depths can be underrepresented in the training dataset which might increase prediction uncertainty | - in 2D kriging soil data (chemical, physical, hydraulic) is first harmonized in training dataset by splining to derive data for fix depth<br><br>- disadvantages: measured soil properties are splined therefore calculated soil hydraulic properties are related to calculated soil properties, thus map relationship between them is derived from interpolated (namely splined) values |
| Spatial inference | - this method relies on the interpolation included in the input layers used for the mapping, thus the mapping is indirect<br><br>- advantage: no further geostatistical analysis is needed to provide 3D information<br><br>- disadvantage: uncertainty of input layers increase uncertainty of predicted soil hydraulic properties | - directly the soil hydraulic properties are interpolated<br><br>- advantage: uncertainty of input layers is decreased due to adding the kriged residuals to the predicted values |
| Information on uncertainty | - interpreted as the uncertainty of the PTFs<br><br>- advantage: can be easily computed for PTFs | - can be derived with e.g. bootstrapping<br><br>- advantages: location specific; the uncertainty accounts for both the unexplained stochastic |

- disadvantages: not location specific, but depends on the input parameter combination, uncertainty of input layers has to be added to the uncertainty of PTFs to provide information on the uncertainty of soil hydraulic maps, uncertainty of input environmental covariates is hardly definable if e.g. 60-70 of them are used for the mapping

variation and the uncertainty in estimating the deterministic model

- disadvantages: computationally demanding; require massive storage capacity; uncertainty of input layers has to be added to the uncertainty of RFK

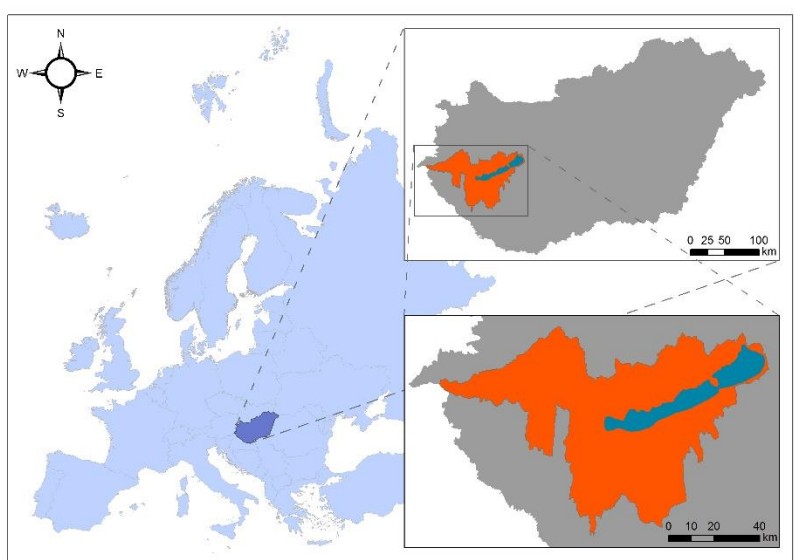

**Figure 1. Location of study site Balaton catchment.**

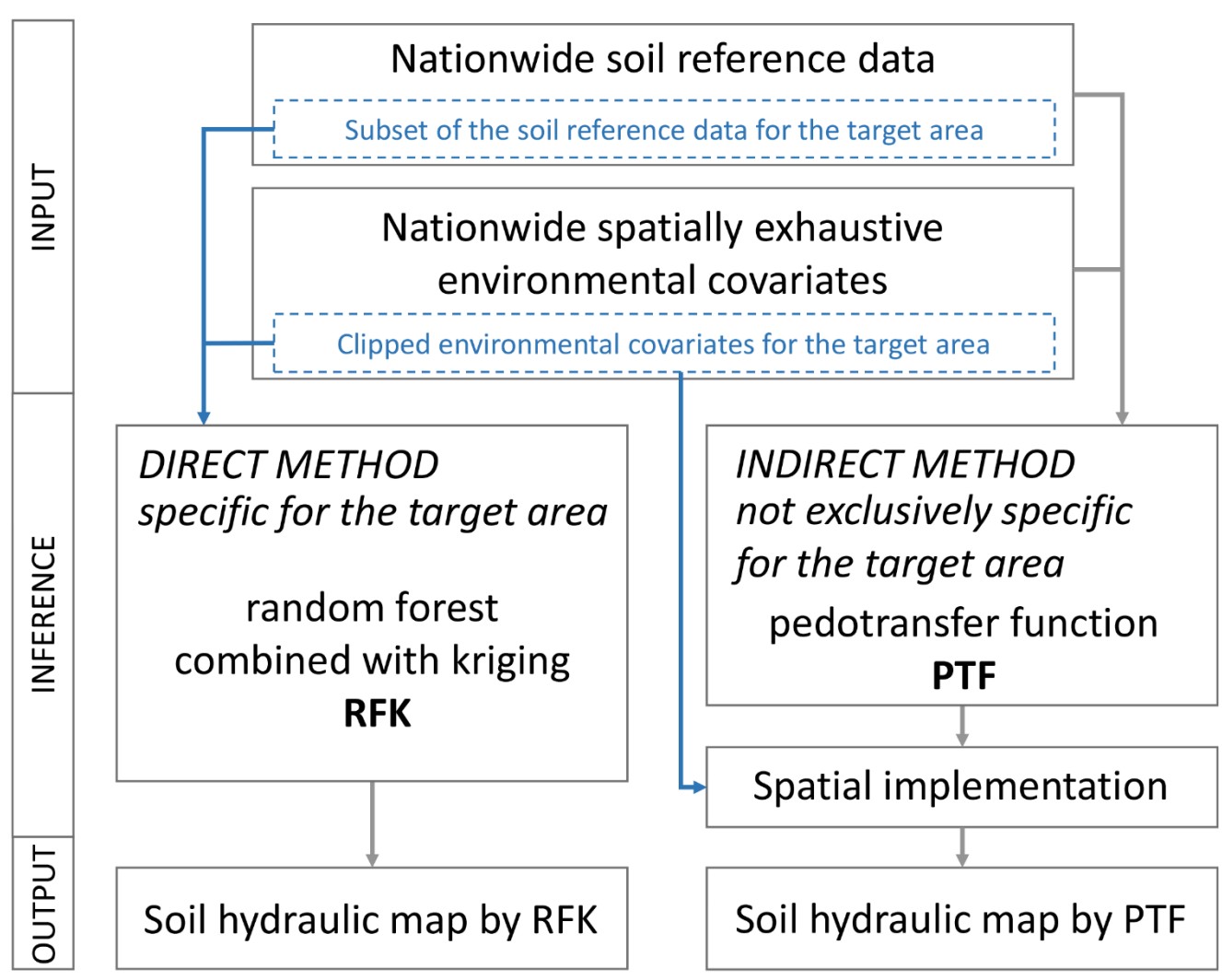

**Figure 2. Flowchart about the main steps of direct and indirect soil hydraulic mapping methods.**

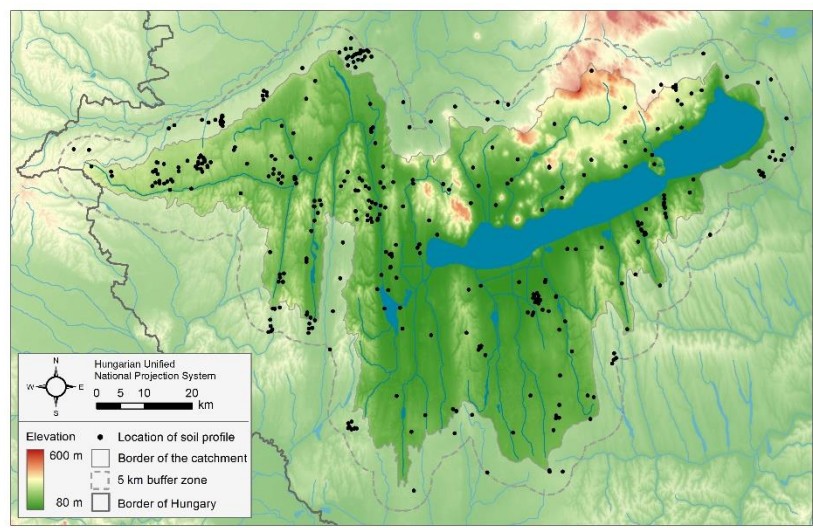

**Figure 3. Location of soil profile used for the geostatistical soil hydraulic mapping on the Balaton catchment study area. Solid line indicates border of the catchment, dashed line shows area with the 5 km buffer zone.**

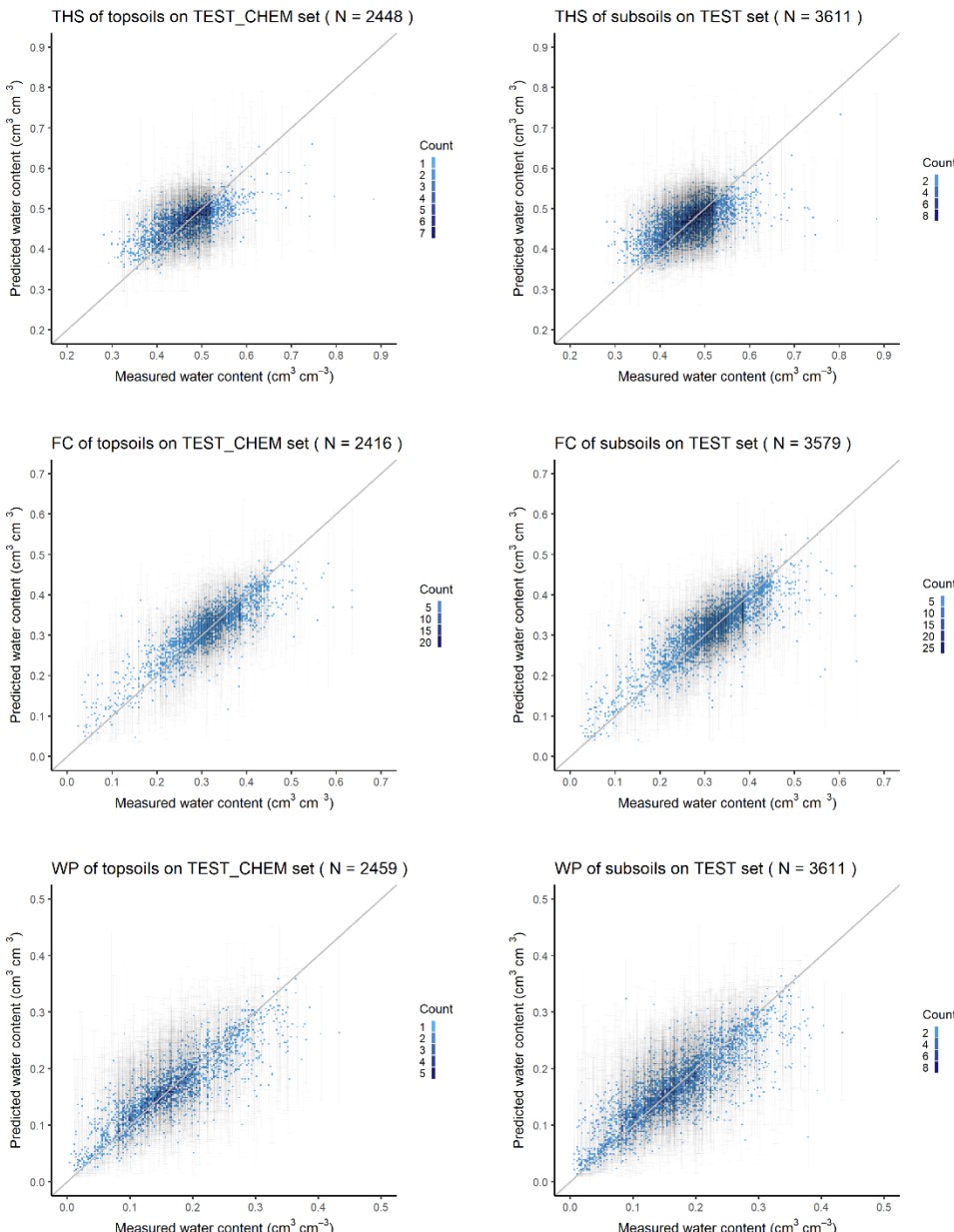

**Figure 4. The scatter plot of the measured versus predicted water retention values with 90% prediction interval on test data sets based on random forest method. THS: saturated water content, FC: water content at field capacity, WP: water content at wilting point, TEST_CHEM set: test dataset in which chemical soil properties are available for the predictions, TEST set: test dataset, in which chemical soil properties are not necessarily available for the predictions.**

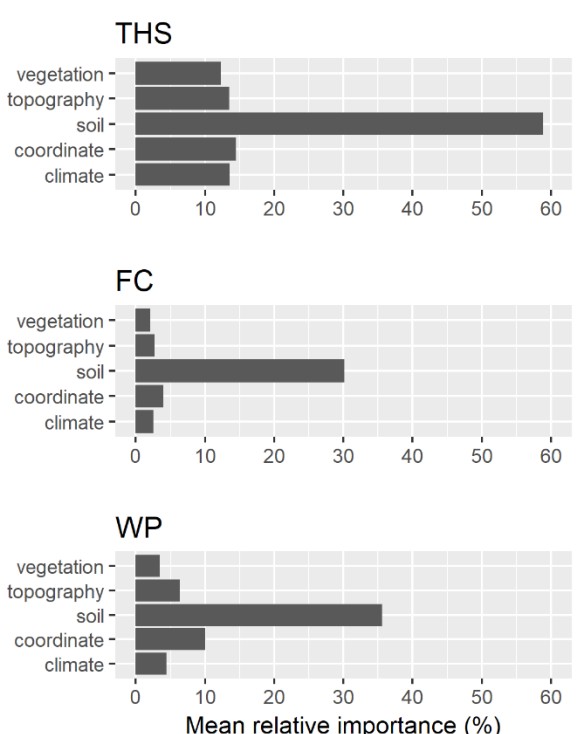

**Figure 5. Mean relative importance of covariates used to predict soil hydraulic properties based on random forest analysis on the training set of MARTHA database. THS: saturated water content, FC: water content at field capacity, WP: water content at wilting point.**

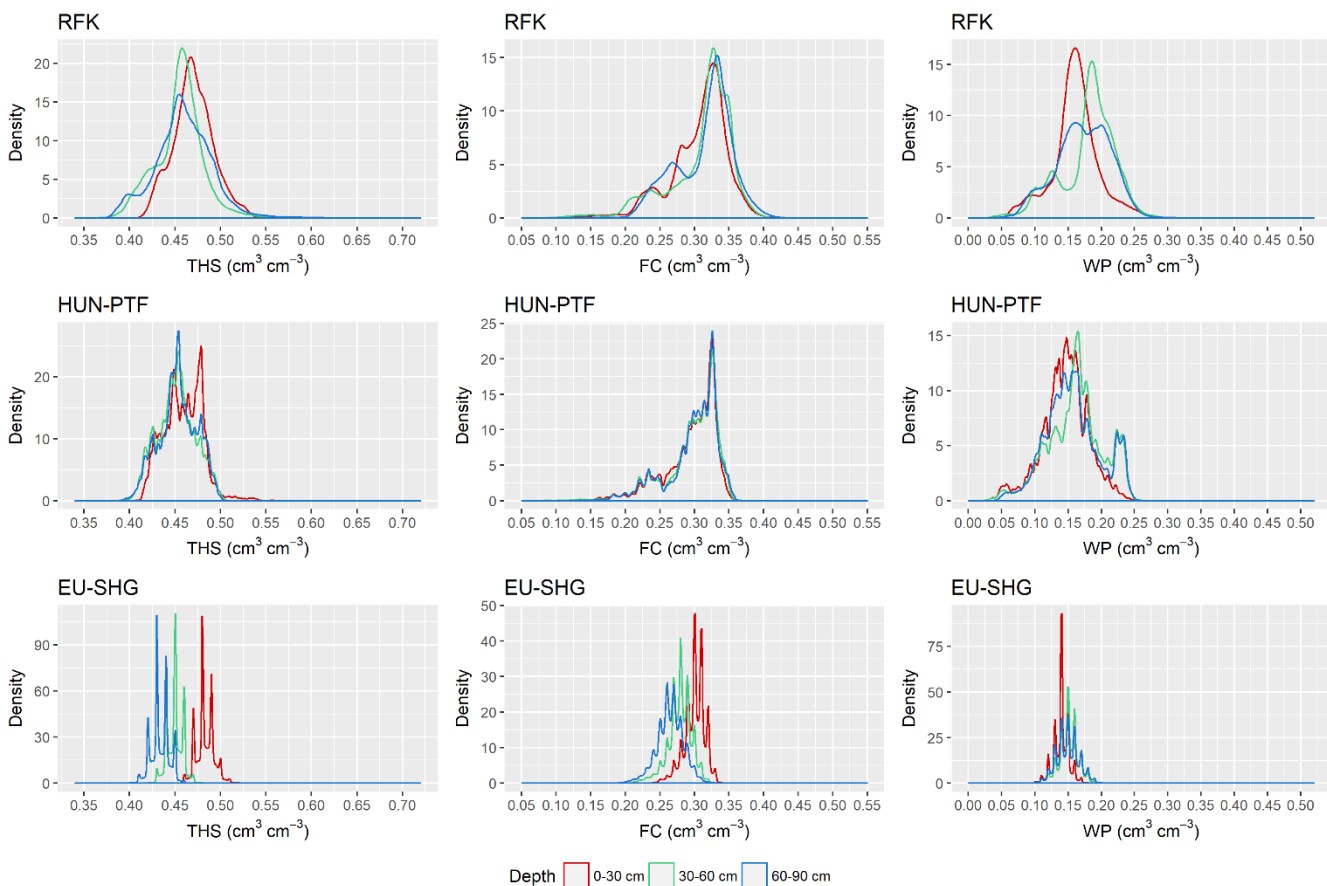

**Figure 6. Density plots of mapped soil hydraulic values by mapping methods and depth. THS: saturated water content, FC: water content at field capacity, WP: water content at wilting point, RFK: derived by random forest with kriging, HUN-PTF: calculated with Hungarian pedotransfer functions, EU-SHG: values from EU-SoilHydroGrids 250m dataset.**

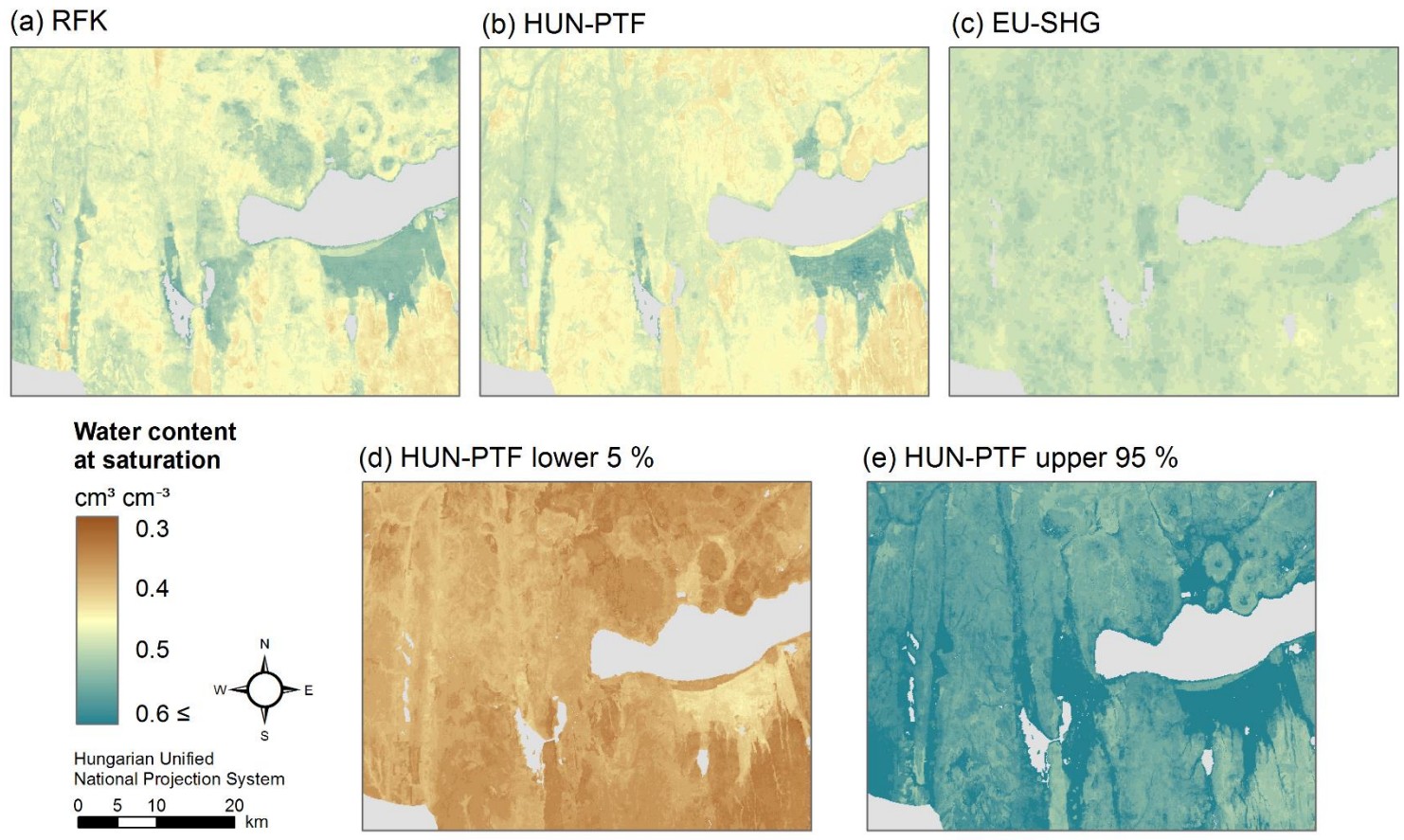

**Figure 7. Map of water content at saturation in 0-30 cm soil depth derived by random forest and kriging mapping approach (RFK) (a), Hungarian pedotransfer functions (HUN-PTF) (b) and cut from the EU-SoilHydroGrids 250m dataset (EU-SHG) (c), possible lower 5 % (d) and upper 95 % (e) based on HUN-PTF for a section of the Balaton catchment.**

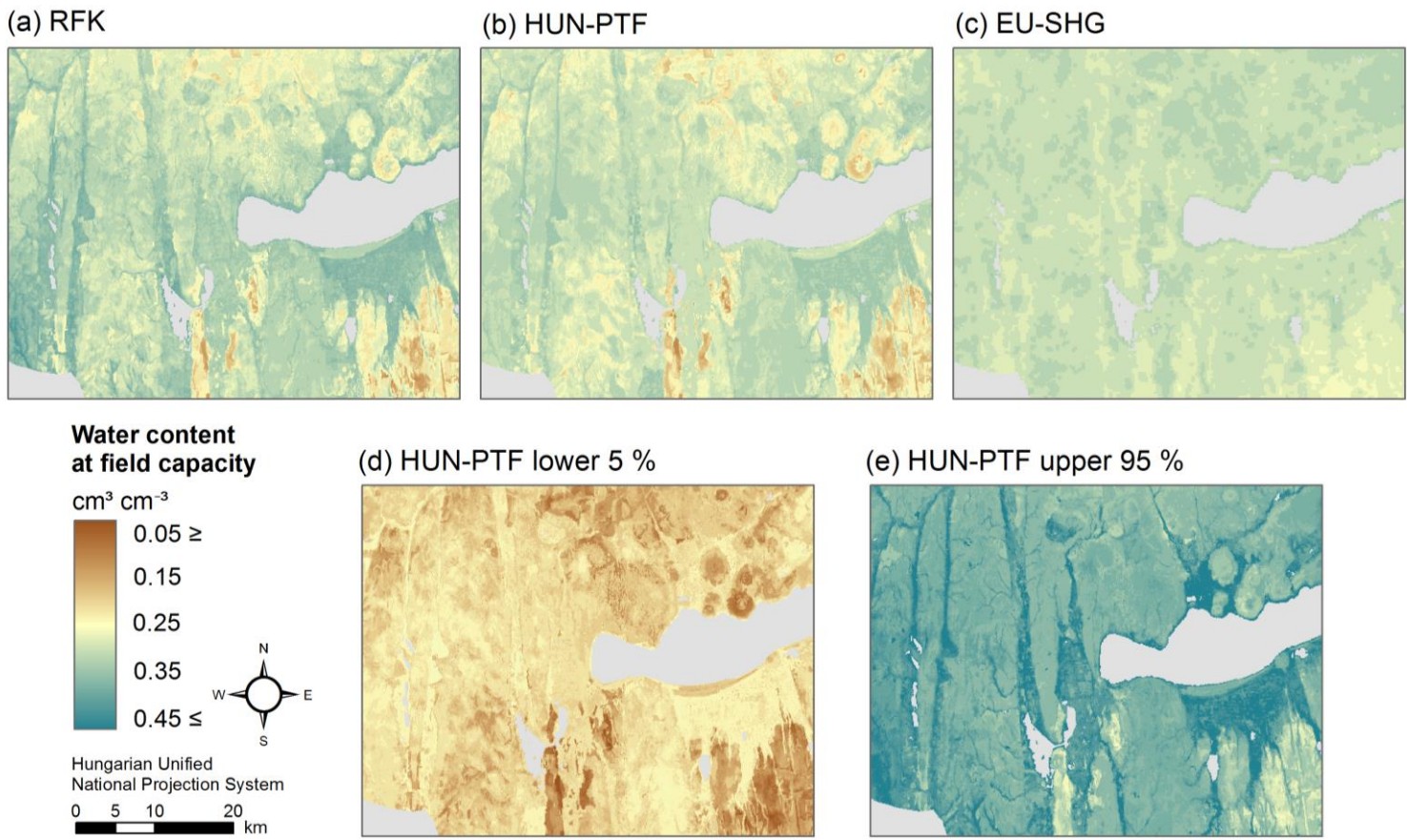

**Figure 8. Map of water content at field capacity in 0-30 cm soil depth derived by random forest and kriging mapping approach (RFK) (a), Hungarian pedotransfer functions (HUN-PTF) (b) and cut from the EU-SoilHydroGrids 250m dataset (EU-SHG) (c), possible lower 5 % (d) and upper 95 % (e) based on HUN-PTF for a section of the Balaton catchment.**

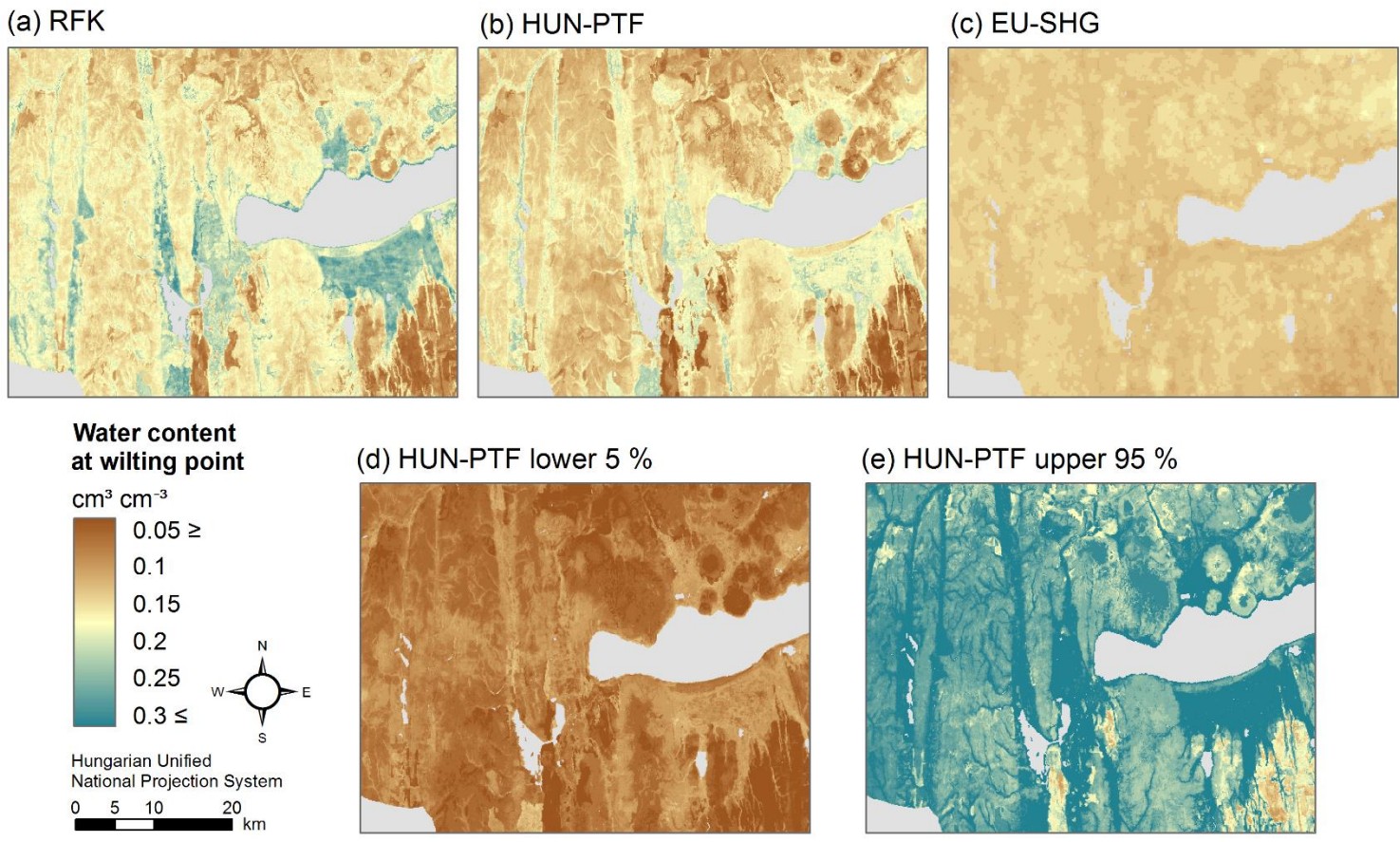

**Figure 9. Map of water content at wilting point in 0-30 cm soil depth derived by random forest and kriging mapping approach (RFK) (a), Hungarian pedotransfer functions (HUN-PTF) (b) and cut from the EU-SoilHydroGrids 250m dataset (EU-SHG) (c), possible lower 5 % (d) and upper 95 % (e) based on HUN-PTF for a section of the Balaton catchment.**

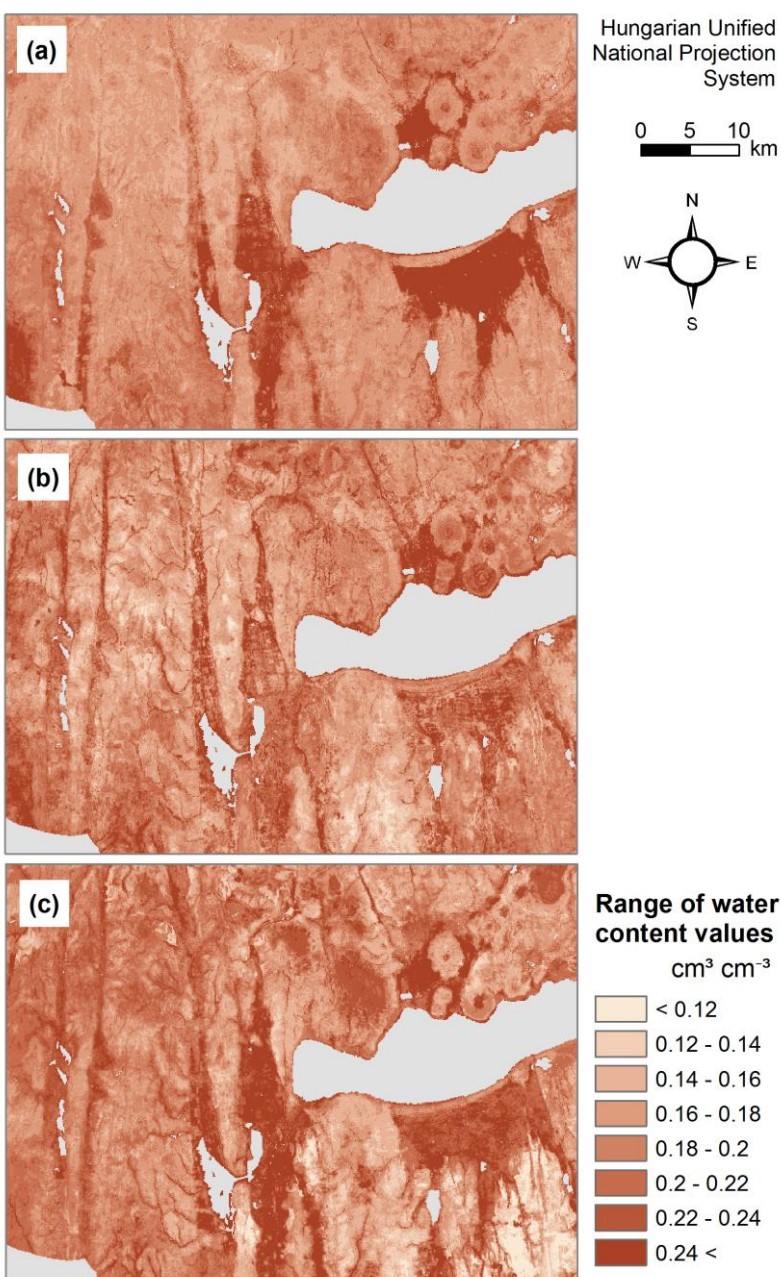

**Figure 10. Differences between possible lower 5 % and upper 95 % water content at saturation (a), field capacity (b) and wilting point (c) in 0-30 cm soil depth for a section of the Balaton catchment.**