# Peer review of "Mapping soil hydraulic properties using random forest based pedotransfer functions and geostatistics"

_Hydrology and Earth System Sciences, 2018_

## Referee Comment (RC1) · Anonymous Referee #1 · 30 Nov 2018

Reviewer comments on "Mapping soil hydraulic properties using random forest based pedotransfer functions" by Brigitta Tóth et al.

Summary In the manuscript by Brigitta Tóth et al. maps of soil water retention characteristics (i.e. soil water contents at saturation, field capacity and the wilting point) are derived for the catchment of lake Balaton from direct measurements (MARTHA data) and additional spatial information on soils, vegetation, topography and climate. In a first step the applicability of two tree-based machine learning algorithms was tested with the result that random forest outperformed generalized boosted regression models. In a second step random forests were combined with classical geostatistical methods to

predict the soil water retention characteristics. However, in most cases the combination of both approaches did not improve the predictions. Resulting maps and pedotransfer functions will be published for non-commercial use.

General comments The study presented in the manuscript is interesting and relevant since spatial information about soil water retention characteristics at regional scale is required for various purposes (e.g. as input data for regional hydrological models or crop modelling). The methods used to predict soil retention characteristics are adequate. However, the procedure of variable selection does not become totally clear. The manuscript is largely well structured, minor changes are suggested in the specific comments below. The conclusion is not in an appropriate form at all and should be written again. I followed the link in the manuscript but could not download the maps and pedotransfer functions. My overall impression is that the work deserves to be published in HESS after major revisions.

Specific comments Abstract P1 L16-17: Please formulate more precise: "water content at saturation (THS), at field capacity (FC), and at the wilting point (WP)"

Introduction P2 L29 – P3 L2: In this paragraph only studies are listed in which tree-based MLA algorithms worked best. Are there also studies where other methods like e.g. artificial neural networks performed best? If yes, they should also be mentioned here. I also think that tree-based methods are a very good choice in this study, but I wonder if there is really only one best approach. P3 L10: Do you mean soil water content at field capacity and wilting point? P3 L11: How can measurements be optimized? What is meant by number of measurements? A large number of? P3 L14-21: Please provide some numbers summarizing the uncertainties found in the studies cited, so the reader can get a feeling about which order of magnitude of uncertainties can be expected when predicting soil retention data. This might also define the "internationally accepted performance of hydraulic PTFs" mentioned in the abstract (P1 L22). P3 L23-25: The objective of the study should be clear and unambiguous. The formulation of the aim(s) should therefore always be identical when mentioned in the text (in

the abstract, in the last paragraph of the introduction and in the first paragraph of the conclusion).

Materials and Methods P4 L12-20: The quite large number of abbreviations introduced in the manuscript unnecessarily demands the capacity of the reader. Please omit abbreviations when the term is used only a few times (e.g. ST or PSD). P4 L22: The covariates are only used to predict the soil hydraulic properties. The relationships between the response and predictor variables are not analysed in the manuscript (e.g. by partial dependence plots). Please rephrase "analysis of the relationships". P4 L22: What does the number 173 stand for? Is it the number of available covariates? P4 L30 – P5 L2: I had to read the sentences several times to understand them. Please rephrase. P4 L25 – P5 L5: The content of the paragraph is not really covered by the heading "Soil hydraulic dataset". Please adapt the heading. I also asked myself, if some information should be shifted to section 2.4.1. P5 L7: Please rephrase "most often used soil water retention values". P5 L7-8: Why did you map water content at -330 cm matric potential when field capacity is determined in Hungary at -300 cm? P5 L15: Why are these methods the most efficient MLAs? This is a very general statement. I am sure that many data scientist would at least partially disagree. Please rephrase. See also my comment on P2 L29 – P3 L2. P5 L17: To calculate quantiles during the predictions? Quantiles of what? What is meant by "during the predictions"? P5 L15-L19: Please add some general information about the principles of regression trees. Also an unexperienced reader should get at least an imagination how the input information is transformed to water retention characteristics. Please also mention once the alternative names of the MLA's (e.g. boosted regression trees) to avoid confusion. P5 L19: . . .build ensembles of models. . . P5 L19: . . . the difference between GBM and RF is the way. . . P5 L26: mtry? Seems to be an argument of an R function? P6 L1: 50 independent variables out of how many? I assume that it is related to the number 173 in P4 L21. Right? P6 L1: It is not really clear to me how you performed the variable selection. Especially when potential predictors are correlated it can be quite challenging to find an optimal set of predictors. Did you start with all possible predictors at once

or did you try out many different combinations of predictors? P6 L8-11: Terminology again: Is it right that "out of bag sampling" is identical to "bootstrapping"? If yes, you might also drop the term "bootstrapping" once. P6 L18: ...to the median and the 5% and 95% quantiles... P6 L20 – P7 L8: The combination of state of the art MLAs and classical geostatistical tools seems plausible and promising to me. However, I wonder if it is correct to call it simply "direct mapping". Isn't it a combination of both: indirect (prediction with RF) and direct (kriging) mapping? Maybe I just haven't understood the essential differences between direct and indirect mapping approaches. P6 L26: ... Table 2 summarizes the measured... P7 L2: Here it says "most important covariates" (the result of the variable selection, right?), but in the caption of Table 1 it says (all) "available environmental covariates". P7 L11: ...with the method... P7 L10: "...based on measured soil hydraulic properties calculated for...". How can the measured properties be calculated? Please rephrase the sentence.

Results and discussion P7 L27-28: In P6 L1 it says that most important 50 independent variables have been selected. How did you select them out of the 69-76 and 65-77 variables mentioned here? P8 L9-16: This paragraph should be shifted to the Materials and Methods section. P8 L6: Why can you assume that multicollinearities are no problem at all? I assume, that many of the predictors presented in Table 1 are highly correlated. I wonder if it is even possible to estimate a unique set of regression-tree parameters when predictors are correlated. For the same reason I could also imagine that it is not possible to determine one unique set of 50 most important independent variables. P8 L22: Please compare the values listed in the text and in Table 1 once again. I am not sure if they match. P8 L27 and many other passages in the text: Is it correct to use the term "covariate" when talking about regression trees? To me "predictors" or "independent variables" seems more plausible. P8 L30: ...than soil related variables... P9 L6-L26: Please explain in the Method section how relative importance is determined. P9 L30: mtry? See also my comment on P5 L26. P9 L27 - P10 L13: In addition to the quality criterions presented in Table 4 it would be interesting to see scatterplots (measured versus predicted values). They sometimes give a better feel-

ing for model performance and they also show if there are areas in the predicted data space of THS, FC and WP with very good or poor prediction performance. P10 L6-13: Please discuss what it is good for to add kriged values computed with a pure nugget model when the residuals of the RF predictions show no spatial structure. This way you simply add random numbers that blur your predicted mean values. I wonder, if you should leave out the whole exercise. P10 L10: the correlation is based on only three pairs of values. Please use a weaker formulation. P10 L33 – P11 L9 and Fig. 5: Why did you select WP in Fig 5 and why did you only show confidence intervals for HUN-PTF? It would also be interesting to see maps of THS and FK and the confidence interval from the RFK predictions. P11 L12: . . .we have not differentiated uncertainty of. . .

Conclusion P12 L1 – P13 L3: The conclusion has poor quality and should be written again. A conclusion should just consist of one or two paragraphs where the most important results are summarized and the most important conclusions are drawn. A concise take home massage can be formulated. In the following just some examples of aspects are listed that are wrong placed the conclusion of the manuscript: P12 L17-20: Such general methodological aspects are not the take home message of the study. P12 L30-32: A discussion of methods or suggestions of alternative methods should be done in the discussion section. P12 L31 – P13 L2: The conclusion is the wrong place for such a detailed discussion of the methods used. A new table (Table 7) should not be introduced in the conclusion section. P13L2-3: A comparison with findings by other authors should be done in the discussion section. New references should not be introduced in the conclusion (e.g. Webster and Oliver (2017)).

---

## Short Comment (SC1) · 21 Dec 2018

Thank you for the detailed review and suggestions which help us to improve our manuscript. We hope to address the comments in a revised version of the article. Below we would like to answer and provide possible solutions for the comments and recommendations, following the referee's questions.

General comments:

Q1. The methods used to predict soil retention characteristics are adequate. However, the procedure of variable selection does not become totally clear.

[Figure]

A: With the variable selection our aim was to exclude less important predictors. Please find detailed description on how variables have been selected under answers for the specific comments.

Q2: The manuscript is largely well structured, minor changes are suggested in the specific comments below. The conclusion is not in an appropriate form at all and should be written again.

A: Thank you for calling our attention to reformat the conclusions. Information will be moved under results and discussion and conclusions will be rewritten. Please find more information on it under answers for specific comments (P12 L1 – P13 L3).

Q3: I followed the link in the manuscript but could not download the maps and pedo-transfer functions.

A: We are sorry that the download link of the maps did not work, something happened with the access authorization after submitting the manuscript, now the problem has been fixed.

Specific comments

Abstract

P1 L16-17: Please formulate more precise: "water content at saturation (THS), at field capacity (FC), and at the wilting point (WP)"

A: Thank you for the suggestion, information on matric potential values will be added in the abstract as well.

Introduction

P2 L29 – P3 L2: In this paragraph only studies are listed in which tree- based MLA algorithms worked best. Are there also studies where other methods like e.g. artificial neural networks performed best? If yes, they should also be mentioned here. I also think that tree-based methods are a very good choice in this study, but I wonder if there

is really only one best approach.

A: Thank you for your comment. We will provide some papers, which used several MLAs (e.g. neural networks, cubist, gradient boosting) for mapping soils and short summary of them.

P3 L10: Do you mean soil water content at field capacity and wilting point?

A: Thank you for highlighting it, yes.

P3 L11: How can measurements be optimized? What is meant by number of measurements? A large number of?

A: You are right, the sampling density was optimized, the sentence will be clarified accordingly.

P3 L14-21: Please provide some numbers summarizing the uncertainties found in the studies cited, so the reader can get a feeling about which order of magnitude of uncertainties can be expected when predicting soil retention data. This might also define the "internationally accepted performance of hydraulic PTFs" mentioned in the abstract (P1 L22).

A: Indeed, it is important, thank you for the idea. In predicting soil water retention data root mean squared error between 0.02 and 0.07 cm3 cm-3 can be expected depending on the predicted soil hydraulic property and available input information (Nguyen et al., 2017; Zhang, Schaap, 2017; Román Dobraco et al., 2019). When PTFs are applied for mapping than uncertainty of the input soil layers will further increase the uncertainty of the PTFs, e.g. in point based validation RMSE was 0.073 cm cm-3 for FC mapped for China in Wu et al. (2018); Leenars et al. (2018) found that mean RMSE for THS, FC and WP together was 0.102 cm3 cm-3 for African soils; in EU-SoilHydroGrids RMSE was 0.095, 0.096, 0.084 cm cm-3 for THS, FC and WP respectively for European samples (Tóth et al., 2017).

Leenaars, J. G. B., Claessens, L., Heuvelink, G. B. M., Hengl, T., Ruiperez González,

M., van Bussel, L. G. J., Guilpart, N., Yang, H. and Cassman, K. G.: Mapping rootable depth and root zone plant-available water holding capacity of the soil of sub-Saharan Africa, Geoderma, 324(February), 18–36, doi:10.1016/j.geoderma.2018.02.046, 2018. Nguyen, P. M., Haghverdi, A., de Pue, J., Botula, Y.-D., Le, K. V., Waegeman, W. and Cornelis, W. M.: Comparison of statistical regression and data-mining techniques in estimating soil water retention of tropical delta soils, Biosyst. Eng., 153, 12–27, doi:10.1016/j.biosystemseng.2016.10.013, 2017. The other references are included in the manuscript.

P3 L23-25: The objective of the study should be clear and unambiguous. The formulation of the aim(s) should therefore always be identical when mentioned in the text (in the abstract, in the last paragraph of the introduction and in the first paragraph of the conclusion).

A: Thank you for the suggestion. We will rephrase the objective mentioned in the text to be identical. We will stick to the following main aim: to analyse difference in performance and spatial patterns between soil hydraulic maps derived with indirect (using PTFs) and direct (geostatistical) mapping methods. The possibility for a non-computation intensive method to map uncertainty of calculated soil hydraulic parameters is a possible advantage of the PTF method.

Materials and Methods

P4 L12-20: The quite large number of abbreviations introduced in the manuscript unnecessarily demands the capacity of the reader. Please omit abbreviations when the term is used only a few times (e.g. ST or PSD).

A: We will remove unnecessary abbreviations.

P4 L22: The covariates are only used to predict the soil hydraulic properties. The relationships be- tween the response and predictor variables are not analysed in the manuscript (e.g. by partial dependence plots). Please rephrase "analysis of the relationships".

A: Thank you for highlighting it, the sentence will be rephrased.

P4 L22: What does the number 173 stand for? Is it the number of available covariates?

A: Yes, we will might delete it to keep the sentence simpler. Number of predictors is mentioned other parts might not be necessary to highlight it also here.

P4 L30 – P5 L2: I had to read the sentences several times to understand them. Please rephrase.

A: We will rephrase those sentences. We needed two kinds of predictions: (1) for topsoils where we could include OM, CaCO3 and pH among the predictors and (2) for subsoils without the above soil chemical parameters, because those are not available for the 30-60 and 60-90 cm on the Balaton catchment. First we randomly selected 67% of the samples from those which has data on the dependent and all the independent variables available on the catchment to derive the PTFs. The rest 33% was used to compare the performance of the PTFs, this we called TEST_CHEM set. In the second step we needed a training (67% of data) and test set (33% of data) also for subsoil prediction for which we didn't have to apply the restriction on the soil chemical properties, therefore we could include more samples for the analysis. As test set we used the samples of the TEST_CHEM set and further added cases to reach the 33% of the complete data appropriate for subsoil predictions. Again the left 67% was used for the training.

P4 L25 – P5 L5: The content of the paragraph is not really covered by the heading "Soil hydraulic dataset". Please adapt the heading. I also asked myself, if some information should be shifted to section 2.4.1.

A: Thank you for the suggestion. Information on how data was selected to train and test the PTFs will be moved under section 2.4.1. The heading will be rephrase, e.g.: Dataset to relate soil hydraulic properties and environmental information.

P5 L7: Please rephrase "most often used soil water retention values".

A: We will rephrase it. We mapped those soil water retention values which are usually differentiated for several applications.

P5 L7-8: Why did you map water content at -330 cm matric potential when field capacity is determined in Hungary at -300 cm?

A: Thank you for finding it, -300 is a mistyping error. It will be corrected in the text.

P5 L15: Why are these methods the most efficient MLAs? This is a very general statement. I am sure that many data scientist would at least partially disagree. Please rephrase. See also my comment on P2 L29 – P3 L2.

A: We will rephrase the sentence, RF and GBM are two widely used MLA, which often achieve good prediction performance on datasets that are characterized by a large number of predictors.

P5 L17: To calculate quantiles during the predictions? Quantiles of what? What is meant by "during the predictions"?

A: The text will be rephrased. "The advantage of these two algorithms is the possibility to estimate prediction intervals as a function of the predictor variables."

P5 L15-L19: Please add some general information about the principles of regression trees. Also an unexperienced reader should get at least an imagination how the input information is transformed to water retention characteristics. Please also mention once the alternative names of the MLA s (e.g. boosted regression trees) to avoid confusion.

A: We will summarize the principles of regression trees before describing the difference between RF and GMB.

P5 L19: . . .build ensembles of models. . .

A: Will be corrected.

P5 L19: . . . the difference between GBM and RF is the way. . .

A: Will be corrected.

P5 L26: mtry? Seems to be an argument of an R function?

A: Yes, we will clarify it in the text.

P6 L1: 50 independent variables out of how many? I assume that it is related to the number 173 in P4 L21. Right?

A: Yes for topsoils, in the case of subsoils it was 170, it will be clarified in the text.

P6 L1: It is not really clear to me how you performed the variable selection. Especially when potential predictors are correlated it can be quite challenging to find an optimal set of predictors. Did you start with all possible predictors at once or did you try out many different combinations of predictors?

A: Thank you for highlighting it, the sentence was not properly phrased. Our aim was to reduce the number of predictors. We selected the 50-50 most important variables both in GBM and RF methods based on the five times repeated five-fold cross-validation, then concatenated the two sets of predictors. In this way less relevant predictors were excluded from the analysis. First we wanted to use the recursive feature elimination (Gregorutti et al., 2017) – with rfe function implemented in R caret package –, which would be a real optimization of input variable selection, but the RFE analysis couldn't be finished on the training set (173 variables of more than 5700 samples) due to lacking computation capacity. Then we found the possibility to at least reducing the number of predictors based on importance measure of the variables. Nussbaum et al. (2018) compared different covariate selection methods: a) based on variable importance calculated in RF model and b) stepwise recursive elimination of the least important variables. They found that both methods selected similar set of covariates. Their study was similar to our ones regarding the topic and dimension of data, therefore based on their results we reduced the number of predictors based on variable importance. , which

is practically the first step of the RFE analysis. In HUN-PTF method we considered the variable importance of both GBM and RF to rely on the results of more methods. We concatenated the 50-50 most important variables, in this way depending on soil hydraulic parameter and soil depth 65-76 predictors stayed in the model. Text is will be modified accordingly.

Gregorutti, B., Michel, B. and Saint-Pierre, P.: Correlation and variable importance in random forests, Stat. Comput., 27(3), 659–678, doi:10.1007/s11222-016-9646-1, 2017.

P6 L8-11: Terminology again: Is it right that "out of bag sampling" is identical to "boot-strapping"? If yes, you might also drop the term "bootstrapping" once.

A: Yes, we will add the term bootstrapping under 2.4.1. section.

P6 L18: . . .to the median and the 5% and 95% quantiles. . .

A: Thank you, we will modify it.

P6 L20 – P7 L8: The combination of state of the art MLAs and classical geostatistical tools seems plausible and promising to me. However, I wonder if it is correct to call it simply "direct mapping". Isn t it a combination of both: indirect (prediction with RF) and direct (kriging) mapping? Maybe I just haven t understood the essential differences between direct and indirect mapping approaches.

A: Thank you for your observation. The essential difference between direct and indirect mapping is the approach of the inference. In direct mapping the target soil variable is directly interpolated over the domain of interest, whereas in indirect mapping not the target variable but its components / factors / covariates are interpolated first and then these interpolated surfaces are in use to compute and map the target variable. Pásztor et al. (2017) discussed this two approaches in detail.

P6 L26: . . . Table 2 summarizes the measured. . .

A: Thank you, we will modify it.

P7 L2: Here it says "most important covariates" (the result of the variable selection, right?), but in the caption of Table 1 it says (all) "available environmental covariates".

A: Thank you for highlighting it. Yes, Table 1 shows all available environmental covariates, text will be corrected.

P7 L11: . . .with the method. . .

A: Thank you, we will modify it.

P7 L10: ". . .based on measured soil hydraulic properties calculated for. . .". How can the measured properties be calculated? Please rephrase the sentence.

A: It will be modified. We intend to say: Performance of soil hydraulic maps was evaluated based on observed soil hydraulic properties harmonized for 0-30, 30-60 and 60-90 cm depth with the method described in 2.4.2 section.

Results and discussion

P7 L27-28: In P6 L1 it says that most important 50 independent variables have been selected. How did you select them out of the 69-76 and 65-77 variables mentioned here?

A: We will add information on how number of variables was decreased under 2.4.1 section. Based on both GBM and RF analysis most important 50-50 variables was selected, after concatenating those we got 69-76 for topsoil predictions, for subsoils 65-77 stayed depending on the target variable in the case of HUN-PTF method.

P8 L9-16: This paragraph should be shifted to the Materials and Methods section.

A: The paragraph includes the result of tuning the model parameters before building the final model, therefore we thought to include it under the results of HUN-PTFs.

P8 L6: Why can you assume that multicollinearities are no problem at all? I assume,

that many of the predictors presented in Table 1 are highly correlated. I wonder if it is even possible to estimate a unique set of regression-tree parameters when predictors are correlated. For the same reason I could also imagine that it is not possible to determine one unique set of 50 most important independent variables.

A: Thanks to highlight it, the sentence will be revised. In case of our analysis multi-collinearity is similar in the training set and mapped area therefore it had less influence on the performance of the maps. It is true that if the HUN-PTFs would be applied in a different region multicollinearities might influence the performance of the predictions. Dorman et al. (2013) found that prediction performance of random forest did not get worse due to high collinearity in the training dataset. The above mentioned RFE analysis would help to decrease multicollinearity (Gregorutti et al., 2017) but didn't run on our dataset due to high dimensionality. By eliminating around 100 predictors from the entire 173 based on the importance measures we could partially decrease the multi-collinearity and improve performance of the prediction. Optimizing predictor selection could be further elaborated, but this is beyond the aim of the presented paper.

Dormann, C. F., Elith, J., Bacher, S., Buchmann, C., Carl, G., Carré, G., Marquéz, J. R. G., Gruber, B., Lafourcade, B., Leitão, P. J., Münkemüller, T., Mcclean, C., Os-borne, P. E., Reineking, B., Schröder, B., Skidmore, A. K., Zurell, D. and Lauten-bach, S.: Collinearity: A review of methods to deal with it and a simulation study evaluating their performance, Ecography (Cop.)., 36(1), 027–046, doi:10.1111/j.1600-0587.2012.07348.x, 2013.

P8 L22: Please compare the values listed in the text and in Table 1 once again. I am not sure if they match.

A: Thank you, you are right, we will correct it. We will modify the R2 values as well, we will list those only for RF. In this way we keep the logic: highlighting results of the selected algorithm.

P8 L27 and many other passages in the text: Is it correct to use the term "covariate"

when talking about regression trees? To me "predictors" or "independent variables" seems more plausible.

A: Thanks for highlighting it. We used the environmental covariates as independent variables in HUN-PTF and the RF part of the RFK. For clarification "predictors" will also be used in text related to PTFs: under 2.1., 2.2, 3.1. Geostatisticians use the term "environmental covariate", therefore it might enhance the interpretability of the manuscript if also this term would be kept.

P8 L30: . . .than soil related variables. . .

A: Thank you, it will be corrected.

P9 L6-L26: Please explain in the Method section how relative importance is determined.

A: Explanation on it will be added.

P9 L30: mtry? See also my comment on P5 L26.

A: We will write it out: "number of randomly selected predictors at each split" for easier understanding.

P9 L27 - P10 L13: In addition to the quality criterions presented in Table 4 it would be interesting to see scatterplots (measured versus predicted values). They sometimes give a better feeling for model performance and they also show if there are areas in the predicted data space of THS, FC and WP with very good or poor prediction performance.

A: Thank you for the suggestion we will include the scatter plots, which would also show the 90 % prediction intervals and describe it in the text. Please find it attached (Fig 4.), quality of figure will be increased: The scatter plot of the measured versus predicted water retention values with 90% prediction interval on test data sets based on random forest method. THS: saturated water content, FC: water content at field capacity, WP:

[Figure]

water content at wilting point (Fig. 4).

P10 L6-13: Please discuss what it is good for to add kriged values computed with a pure nugget model when the residuals of the RF predictions show no spatial structure. This way you simply add random numbers that blur your predicted mean values. I wonder, if you should leave out the whole exercise.

A: You are right in that sense kriged values computed with a pure nugget model do not give any new "information" to the RF predictions. However, kriged values with a nugget variogram add zero values to the RF predictions rather than random numbers. Thus kriging with a nugget model do not blur the predictions. We wouldn't like to leave this exercise because it is an algorithmical decision (considering the stochastic part of the spatial variation of the given soil property) rather than a subjective decision, even if we get the same result.

P10 L10: the correlation is based on only three pairs of values. Please use a weaker formulation.

A: Thank you, we will rephrase the sentence.

P10 L33 – P11 L9 and Fig. 5: Why did you select WP in Fig 5 and why did you only show confidence intervals for HUN-PTF? It would also be interesting to see maps of THS and FK and the confidence interval from the RFK predictions.

A: We will show the Fig. 5 and 6 maps also for THS and FC, please find those attached (Fig. 1-3). Sorry for confusion, the formulation of the aim of the paper will be identical in the entire manuscript. Calculating the confidence intervals for the RFK method is beyond the scope of this study, although it would be interesting to analyze the difference between uncertainty maps calculated with the different methods in the future, similarly as it was done by Szatmári and Pásztor (2018) for soil organic carbon stock in Hungary. According to it, quantile regression forest (Meinshausen, 2006) based uncertainty quantification outperforms most of the prediction techniques used in digital soil

mapping. Furthermore, they have pointed out that bootstrapping based uncertainty quantification for RFK is quite time consuming, as well as it requires massive storage and computing capacity. The ranger package - with which we derived the HUN-PTFs - includes implementation of quantile regression forest (Meinshausen, 2006) for the calculations of the prediction intervals.

Meinshausen, N.: Quantile Regression Forests, J. Mach. Learn. Res., 7, 983–999, 2006.

P11 L12: . . .we have not differentiated uncertainty of. . .

A: Thank you, it will be corrected.

Conclusion

P12 L1 – P13 L3: The conclusion has poor quality and should be written again. A conclusion should just consist of one or two paragraphs where the most important results are summarized and the most important conclusions are drawn. A concise take home massage can be formulated. In the following just some examples of aspects are listed that are wrong placed the conclusion of the manuscript: P12 L17- 20: Such general methodological aspects are not the take home message of the study.

A: We were not aware of the correct formulation of the conclusions and wrongly included discussion in that section. The text will be completely moved under Results and discussion section and will add the real conclusions there. The take home message is the following: Based on the results in case of six out of nine soil hydraulic maps there is no significant difference in performance between values derived with pedotransfer function and geostatistical method on the Balaton catchment. The benefit of maps derived with random forest and kriging is that locally extreme values can be better characterized. In the case of pedotransfer function based mapping it is advantageous that calculation of uncertainty is much less computation intensive than it is with geostatistical methods, although it would be interesting in the future to analyze the

difference between uncertainty maps calculated with the different methods specifically for soil hydraulic properties.

P12 L30-32: A discussion of methods or suggestions of alternative methods should be done in the discussion section.

A: It will be moved under Results and discussion section.

P12 L31 – P13 L2: The conclusion is the wrong place for such a detailed discussion of the methods used. A new table (Table 7) should not be introduced in the conclusion section.

A: Both text and Table 7 will be moved under Results and discussion section.

P13L2-3: A comparison with findings by other authors should be done in the discussion section. New references should not be introduced in the conclusion (e.g. Webster and Oliver (2017)).

A: Text will be moved under Results and discussion section.

Thank you again the comments and suggestions. We hope to adequately address the issues identified and look forward to any other feedback the referee may have.
* * *
[Figure]

**Fig. 1.** THS_map_section

[Figure]

Water content
at field capacity

cm³ cm⁻³
0.05 ≥
0.15
0.25
0.35
0.45 ≤

Hungarian Unified
National Projection System
0  5  10    20
             km

(a) RFK
(b) HUN-PTF
(c) EU-SHG
(d) HUN-PTF lower 5 %
(e) HUN-PTF upper 95 %

**Fig. 2.** FC_map_section

[Figure]

**Fig. 3.** Difference between possible lower 5 % and upper 95 % water content at saturation (a), field capacity (b) and wilting point (c) in 0-30 cm soil depth for a section of the Balaton catchment.

THS of topsoils on TEST_CHEM set (N = 2448 )

THS of subsoils on TEST set (N = 3611 )

FC of topsoils on TEST_CHEM set (N = 2416 )

FC of subsoils on TEST set (N = 3579 )

WP of topsoils on TEST_CHEM set (N = 2459 )

WP of subsoils on TEST set (N = 3611 )

**Fig. 4.** The scatter plot of the measured versus predicted water retention values with 90% prediction interval on test data sets based on random forest method.

---

## Referee Comment (RC2) · Anonymous Referee #2 · 31 Jan 2019

This is an interesting manuscript investigating an important topic. The manuscript is well structured, but it is difficult to follow in places. Especially the methods need further clarification and details.

Lines 11-13: very long sentence Lines 13-16: not clear

Abstract: in general: a lot of acronyms for an abstract. Not clear and difficult to understand what direct and indirect are.

Introduction: I think a language revision is needed. The first paragraph for example is difficult to understand.

[Figure]

Lines 19-22: this is not always true. Please discuss.

Section 2.2: the description of the soil hydraulic dataset is not clear. The split between test and training in particular.

The results are well presented, but there are minor (and less minor) problems with typos and structure of the sentences.

---

## Author Comment (AC1) · 11 Feb 2019

Reviewer comments on "Mapping soil hydraulic properties using random forest based pedotransfer functions" by Brigitta Tóth et al.

Summary

In the manuscript by Brigitta Tóth et al. maps of soil water retention characteristics (i.e.

[Figure]

soil water contents at saturation, field capacity and the wilting point) are derived for the catchment of lake Balaton from direct measurements (MARTHA data) and additional spatial information on soils, vegetation, topography and climate. In a first step the applicability of two tree-based machine learning algorithms was tested with the result that random forest outperformed generalized boosted regression models. In a second step random forests were combined with classical geostatistical methods to predict the soil water retention characteristics. However, in most cases the combination of both approaches did not improve the predictions. Resulting maps and pedotransfer functions will be published for non-commercial use.

General comments

The study presented in the manuscript is interesting and relevant since spatial information about soil water retention characteristics at regional scale is required for various purposes (e.g. as input data for regional hydrological models or crop modelling). The methods used to predict soil retention characteristics are adequate. However, the procedure of variable selection does not become totally clear. The manuscript is largely well structured, minor changes are suggested in the specific comments below. The conclusion is not in an appropriate form at all and should be written again. I followed the link in the manuscript but could not download the maps and pedotransfer functions. My overall impression is that the work deserves to be published in HESS after major revisions.

A: Thank you for the detailed review and suggestions which help us to improve our manuscript. We hope to address the comments in a revised version of the article. Below we would like to answer and provide possible solutions for the comments and recommendations, following the referee's questions.

General comments:

Q1. The methods used to predict soil retention characteristics are adequate. However, the procedure of variable selection does not become totally clear.

A: With the variable selection our aim was to exclude less important predictors. Please find detailed description on how variables have been selected under answers for the specific comments.

Q2: The manuscript is largely well structured, minor changes are suggested in the specific comments below. The conclusion is not in an appropriate form at all and should be written again.

A: Thank you for calling our attention to reformat the conclusions. Information will be moved under results and discussion and conclusions will be rewritten. Please find more information on it under answers for specific comments (P12 L1 – P13 L3).

Q3: I followed the link in the manuscript but could not download the maps and pedo-transfer functions.

A: We are sorry that the download link of the maps did not work, something happened with the access authorization after submitting the manuscript, now the problem has been fixed.

Specific comments

Abstract

P1 L16-17: Please formulate more precise: "water content at saturation (THS), at field capacity (FC), and at the wilting point (WP)"

A: Thank you for the suggestion, information on matric potential values will be added in the abstract as well.

Introduction

P2 L29 – P3 L2: In this paragraph only studies are listed in which tree- based MLA algorithms worked best. Are there also studies where other methods like e.g. artificial neural networks performed best? If yes, they should also be mentioned here. I also think that tree-based methods are a very good choice in this study, but I wonder if there

is really only one best approach.

A: Thank you for your comment. We will provide some papers, which used several MLAs (e.g. neural networks, cubist, gradient boosting) for mapping soils and short summary of them: "Adhikari et al. (2014) used cubist combined with kriging for mapping soil organic carbon concentration and stock in Denmark and they found that cubist was appropriate for this purpose. The same was observed by Matos-Moreira et al. (2017), they used cubist for mapping the phosphorus concentration in north-western France. Behrens et al. (2018) compared a number of state of the art digital soil mapping methods including geostatistical techniques (i.e. ordinary kriging, regression kriging and geographically weighted regression) and machine learning algorithms (i.e. multivariate adaptive regression splines, radial basis function support vector machines, cubist, random forest and neural networks). They obtained the best results with cubist, random forest and bagged multivariate adaptive regression splines."

P3 L10: Do you mean soil water content at field capacity and wilting point?

A: Thank you for highlighting it, yes.

P3 L11: How can measurements be optimized? What is meant by number of measurements? A large number of?

A: You are right, the sampling density was optimized, the sentence will be clarified accordingly.

P3 L14-21: Please provide some numbers summarizing the uncertainties found in the studies cited, so the reader can get a feeling about which order of magnitude of uncertainties can be expected when predicting soil retention data. This might also define the "internationally accepted performance of hydraulic PTFs" mentioned in the abstract (P1 L22).

A: Indeed, it is important, thank you for the idea. In predicting soil water retention data root mean squared error between 0.02 and 0.07 cm3 cm-3 can be expected depending

on the predicted soil hydraulic property and available input information (Nguyen et al., 2017; Zhang, Schaap, 2017; Román Dobraco et al., 2019). When PTFs are applied for mapping then uncertainty of the input soil layers will further increase the uncertainty of the PTFs, e.g. in point based validation RMSE was 0.073 cm cm-3 for FC mapped for China in Wu et al. (2018); Leenars et al. (2018) found that mean RMSE for THS, FC and WP together was 0.102 cm3 cm-3 for African soils; in EU-SoilHydroGrids RMSE was 0.095, 0.096, 0.084 cm cm-3 for THS, FC and WP respectively for European samples (Tóth et al., 2017). We will add this information in the text with the following references.

Leenaars, J. G. B., Claessens, L., Heuvelink, G. B. M., Hengl, T., Ruiperez González, M., van Bussel, L. G. J., Guilpart, N., Yang, H. and Cassman, K. G.: Mapping rootable depth and root zone plant-available water holding capacity of the soil of sub-Saharan Africa, Geoderma, 324(February), 18–36, doi:10.1016/j.geoderma.2018.02.046, 2018. Nguyen, P. M., Haghverdi, A., de Pue, J., Botula, Y.-D., Le, K. V., Waegeman, W. and Cornelis, W. M.: Comparison of statistical regression and data-mining techniques in estimating soil water retention of tropical delta soils, Biosyst. Eng., 153, 12–27, doi:10.1016/j.biosystemseng.2016.10.013, 2017.

P3 L23-25: The objective of the study should be clear and unambiguous. The formulation of the aim(s) should therefore always be identical when mentioned in the text (in the abstract, in the last paragraph of the introduction and in the first paragraph of the conclusion).

A: Thank you for the suggestion. We will rephrase the objective mentioned in the text to be identical. We will stick to the following main aim: to analyse difference in performance and spatial patterns between soil hydraulic maps derived with indirect (using PTFs) and direct (geostatistical) mapping methods. The possibility for a non-computation intensive method to map uncertainty of calculated soil hydraulic parameters is a possible advantage of the PTF method.

Materials and Methods

P4 L12-20: The quite large number of abbreviations introduced in the manuscript unnecessarily demands the capacity of the reader. Please omit abbreviations when the term is used only a few times (e.g. ST or PSD).

A: We will decrease the number of abbreviations, we will not use MLA, OM, CaCO3 and EU-SHG in the text and will also remove unnecessary ST and PSD abbreviations. In the abstract we will not introduce the THS, FC, WP and MARTHA abbreviations.

P4 L22: The covariates are only used to predict the soil hydraulic properties. The relationships be- tween the response and predictor variables are not analysed in the manuscript (e.g. by partial dependence plots). Please rephrase "analysis of the relationships".

A: Thank you for highlighting it, the sentence will be rephrased: "For the prediction of soil hydraulic properties based on soil and other environmental variables we used . . ."

P4 L22: What does the number 173 stand for? Is it the number of available covariates?

A: Yes, we might delete it to keep the sentence simpler. Number of predictors is mentioned elsewhere, it might not be necessary to highlight it also here.

P4 L30 – P5 L2: I had to read the sentences several times to understand them. Please rephrase.

A: We will rephrase those sentences: "For the construction of PTFs those samples were selected from the MARTHA dataset which have measured information on dependent and independent variables. We needed two kinds of predictions: (1) for topsoils where we could include organic matter content, calcium carbonate content and pH among the predictors and (2) for subsoils without the above soil chemical parameters, because those are not available for the 30-60 and 60-90 cm soil depths on the Balaton catchment. First we randomly selected 67% of the samples from those which have data on the dependent and all the independent variables available on the catchment

area to derive the PTFs. The rest 33% was used to compare the performance of the PTFs, this we called TEST_CHEM set. In the second step we needed a training (67% of data) and test set (33% of data) also for subsoil prediction for which we didn't have to apply the restriction on the soil chemical properties, therefore we could include more samples for the analysis. As test set we used the samples of the TEST_CHEM set and further added cases to reach the 33% of the complete data appropriate for subsoil predictions. Again the left 67% was used for the training. Number of samples used to train and test the PTFs was 8157 and 12039 for THS, 8051 and 11931 for FC, 8195 and 12036 for WP, with and without soil chemical properties respectively."

P4 L25 – P5 L5: The content of the paragraph is not really covered by the heading "Soil hydraulic dataset". Please adapt the heading. I also asked myself, if some information should be shifted to section 2.4.1.

A: Thank you for the suggestion. Information on how data was selected to train and test the PTFs will be moved under section 2.4.1. The heading will be rephrase, e.g.: Dataset to relate soil hydraulic properties and environmental information.

P5 L7: Please rephrase "most often used soil water retention values".

A: We will rephrase it. We mapped those soil water retention values which are usually differentiated for several applications.

P5 L7-8: Why did you map water content at -330 cm matric potential when field capacity is determined in Hungary at -300 cm?

A: Thank you for finding it, -300 is a mistyping error. It will be corrected in the text.

P5 L15: Why are these methods the most efficient MLAs? This is a very general statement. I am sure that many data scientist would at least partially disagree. Please rephrase. See also my comment on P2 L29 – P3 L2.

A: We will rephrase the sentence, RF and GBM are two widely used MLA, which often achieve good prediction performance on datasets that are characterized by a large

number of predictors.

P5 L17: To calculate quantiles during the predictions? Quantiles of what? What is meant by "during the predictions"?

A: The text will be rephrased. "The advantage of these two algorithms is the possibility to estimate prediction intervals of the dependent variable as a function of the independent variables."

P5 L15-L19: Please add some general information about the principles of regression trees. Also an unexperienced reader should get at least an imagination how the input information is transformed to water retention characteristics. Please also mention once the alternative names of the MLA s (e.g. boosted regression trees) to avoid confusion.

A: We will summarize the principles of regression trees before describing the difference between RF and GMB.

P5 L19: . . .build ensembles of models. . .

A: Will be corrected.

P5 L19: . . . the difference between GBM and RF is the way. . .

A: Will be corrected.

P5 L26: mtry? Seems to be an argument of an R function?

A: Yes, we will clarify it in the text.

P6 L1: 50 independent variables out of how many? I assume that it is related to the number 173 in P4 L21. Right?

A: Yes for topsoils, in the case of subsoils it was 170, it will be clarified in the text.

P6 L1: It is not really clear to me how you performed the variable selection. Especially when potential predictors are correlated it can be quite challenging to find an optimal set of predictors. Did you start with all possible predictors at once or did you try out

many different combinations of predictors?

A: Thank you for highlighting it, the sentence was not properly phrased. Our aim was to reduce the number of predictors. We selected the 50-50 most important variables both in GBM and RF methods based on the five times repeated five-fold cross-validation, then concatenated the two sets of predictors. In this way less relevant predictors were excluded from the analysis. First we wanted to use the recursive feature elimination (Gregorutti et al., 2017) – with rfe function implemented in R caret package –, which would be a real optimization of input variable selection, but the RFE analysis couldn't be finished on the training set (173 variables of more than 5700 samples) due to lacking computation capacity. Then we found the possibility to at least reducing the number of predictors based on importance measure of the variables. Nussbaum et al. (2018) compared different covariate selection methods: a) based on variable importance calculated in RF model and b) stepwise recursive elimination of the least important variables. They found that both methods selected similar set of covariates. Their study was similar to our ones regarding the topic and dimension of data, therefore based on their results we reduced the number of predictors based on variable importance, which is practically the first step of the RFE analysis. In HUN-PTF method we considered the variable importance of both GBM and RF to rely on the results of two different methods. We concatenated the 50-50 most important variables, in this way depending on soil hydraulic parameter and soil depth 65-76 predictors stayed in the model. Text will be modified accordingly.

Gregorutti, B., Michel, B. and Saint-Pierre, P.: Correlation and variable importance in random forests, Stat. Comput., 27(3), 659–678, doi:10.1007/s11222-016-9646-1, 2017.

P6 L8-11: Terminology again: Is it right that "out of bag sampling" is identical to "bootstrapping"? If yes, you might also drop the term "bootstrapping" once.

A: Yes, we will add the term bootstrapping under 2.4.1. section.

[Figure]

P6 L18: . . .to the median and the 5% and 95% quantiles. . .

A: Thank you, we will modify it.

P6 L20 – P7 L8: The combination of state of the art MLAs and classical geostatistical tools seems plausible and promising to me. However, I wonder if it is correct to call it simply "direct mapping". Isn t it a combination of both: indirect (prediction with RF) and direct (kriging) mapping? Maybe I just haven t understood the essential differences between direct and indirect mapping approaches.

A: Thank you for your observation. The essential difference between direct and indirect mapping is the approach of the inference. In direct mapping the target soil variable is directly interpolated over the domain of interest, whereas in indirect mapping not the target variable but its components / factors / covariates are interpolated first and then these interpolated surfaces are in use to compute and map the target variable. Pásztor et al. (2017) discussed this two approaches in detail.

P6 L26: . . . Table 2 summarizes the measured. . .

A: Thank you, we will modify it.

P7 L2: Here it says "most important covariates" (the result of the variable selection, right?), but in the caption of Table 1 it says (all) "available environmental covariates".

A: Thank you for highlighting it. Yes, Table 1 shows all available environmental covariates, text will be corrected.

P7 L11: . . .with the method. . .

A: Thank you, we will modify it.

P7 L10: ". . .based on measured soil hydraulic properties calculated for. . .". How can the measured properties be calculated? Please rephrase the sentence.

A: It will be modified. We intend to say: Performance of soil hydraulic maps was evaluated based on observed soil hydraulic properties harmonized for 0-30, 30-60 and 60-90 cm depth with the method described in 2.4.2 section.

Results and discussion

P7 L27-28: In P6 L1 it says that most important 50 independent variables have been selected. How did you select them out of the 69-76 and 65-77 variables mentioned here?

A: We will add information on how number of variables was decreased under 2.4.1 section. Based on both GBM and RF analysis most important 50-50 variables was selected, after concatenating those we got 69-76 for topsoil predictions, for subsoils 65-77 stayed depending on the target variable in the case of HUN-PTF method.

P8 L9-16: This paragraph should be shifted to the Materials and Methods section.

A: The paragraph includes the result of tuning the model parameters before building the final model, therefore we thought to include it under the results of HUN-PTFs.

P8 L6: Why can you assume that multicollinearities are no problem at all? I assume, that many of the predictors presented in Table 1 are highly correlated. I wonder if it is even possible to estimate a unique set of regression-tree parameters when predictors are correlated. For the same reason I could also imagine that it is not possible to determine one unique set of 50 most important independent variables.

A: Thanks to highlight it, the sentence will be revised. In case of our analysis multi-collinearity is similar in the training set and mapped area therefore it had less influence on the performance of the maps. It is true that if the HUN-PTFs would be applied in a different region multicollinearities might influence the performance of the predictions. Dorman et al. (2013) found that prediction performance of random forest did not get worse due to high collinearity in the training dataset. The above mentioned RFE analysis would help to decrease multicollinearity (Gregorutti et al., 2017) but didn't run on our dataset due to high dimensionality. By eliminating around 100 predictors from the

entire 173 based on the importance measures we could partially decrease the multi-collinearity and improve performance of the prediction. Optimizing predictor selection could be further elaborated, but this is beyond the aim of the presented paper.

Dormann, C. F., Elith, J., Bacher, S., Buchmann, C., Carl, G., Carré, G., Marquéz, J. R. G., Gruber, B., Lafourcade, B., Leitão, P. J., Münkemüller, T., Mcclean, C., Osborne, P. E., Reineking, B., Schröder, B., Skidmore, A. K., Zurell, D. and Lautenbach, S.: Collinearity: A review of methods to deal with it and a simulation study evaluating their performance, Ecography (Cop.)., 36(1), 027–046, doi:10.1111/j.1600-0587.2012.07348.x, 2013.

P8 L22: Please compare the values listed in the text and in Table 1 once again. I am not sure if they match.

A: Thank you, you are right, we will correct it. We will modify the R2 values as well, we will list those only for RF. In this way we keep the logic: highlighting results of the selected algorithm.

P8 L27 and many other passages in the text: Is it correct to use the term "covariate" when talking about regression trees? To me "predictors" or "independent variables" seems more plausible.

A: Thanks for highlighting it. We used the environmental covariates as independent variables in HUN-PTF and the RF part of the RFK. For clarification "predictors" will also be used in text related to PTFs: under 2.1., 2.2, 3.1. Geostatisticians use the term "environmental covariate", therefore it might enhance the interpretability of the manuscript if also this term would be kept.

P8 L30: . . .than soil related variables. . .

A: Thank you, it will be corrected.

P9 L6-L26: Please explain in the Method section how relative importance is determined.

A: Explanation on it will be added.

P9 L30: mtry? See also my comment on P5 L26.

A: We will write it out: "number of randomly selected predictors at each split" for easier understanding.

P9 L27 - P10 L13: In addition to the quality criterions presented in Table 4 it would be interesting to see scatterplots (measured versus predicted values). They sometimes give a better feeling for model performance and they also show if there are areas in the predicted data space of THS, FC and WP with very good or poor prediction performance.

A: Thank you for the suggestion we will include the scatter plots after Fig. 2, which would also show the 90 % prediction intervals and describe it in the text. Please find it attached as P1., quality of figure will be increased: The scatter plot of the measured versus predicted water retention values with 90% prediction interval on test data sets based on random forest method. THS: saturated water content, FC: water content at field capacity, WP: water content at wilting point.

P10 L6-13: Please discuss what it is good for to add kriged values computed with a pure nugget model when the residuals of the RF predictions show no spatial structure. This way you simply add random numbers that blur your predicted mean values. I wonder, if you should leave out the whole exercise.

A: You are right in that sense kriged values computed with a pure nugget model do not give any new "information" to the RF predictions. However, kriged values with a nugget variogram add zero values to the RF predictions rather than random numbers. Thus kriging with a nugget model do not blur the predictions. We would not like to leave this exercise because it is an algorithmical decision (considering the stochastic part of the spatial variation of the given soil property) rather than a subjective decision, even if we get the same result.
P10 L10: the correlation is based on only three pairs of values. Please use a weaker formulation.

A: Thank you, we will rephrase the sentence.

P10 L33 – P11 L9 and Fig. 5: Why did you select WP in Fig 5 and why did you only show confidence intervals for HUN-PTF? It would also be interesting to see maps of THS and FK and the confidence interval from the RFK predictions.

A: We will show the Fig. 5 and 6 maps also for THS and FC, please find those attached (P2, P3, P4). Sorry for confusion, the formulation of the aim of the paper will be identical in the entire manuscript. Calculating the confidence intervals for the RFK method is beyond the scope of this study, although it would be interesting to analyze the difference between uncertainty maps calculated with the different methods in the future, similarly as it was done by Szatmári and Pásztor (2018) for soil organic carbon stock in Hungary. According to it, quantile regression forest (Meinshausen, 2006) based uncertainty quantification outperforms most of the prediction techniques used in digital soil mapping. Furthermore, they have pointed out that bootstrapping based uncertainty quantification for RFK is quite time consuming, as well as it requires massive storage and computing capacity. The ranger package - with which we derived the HUN-PTFs - includes implementation of quantile regression forest (Meinshausen, 2006) for the calculations of the prediction intervals.

Meinshausen, N.: Quantile Regression Forests, J. Mach. Learn. Res., 7, 983–999, 2006.

P11 L12: . . .we have not differentiated uncertainty of. . .

A: Thank you, it will be corrected.

Conclusion

P12 L1 – P13 L3: The conclusion has poor quality and should be written again. A conclusion should just consist of one or two paragraphs where the most important

results are summarized and the most important conclusions are drawn. A concise take home massage can be formulated. In the following just some examples of aspects are listed that are wrong placed the conclusion of the manuscript: P12 L17- 20: Such general methodological aspects are not the take home message of the study.

A: We were not aware of the correct formulation of the conclusions and wrongly included discussion in that section. The text will be completely moved under Results and discussion section and we will add the real conclusions there. The take home message is the following: Based on the results in case of six out of nine soil hydraulic maps there is no significant difference in performance between values derived with pedotransfer function and geostatistical method on the Balaton catchment. The benefit of maps derived with random forest and kriging is that locally extreme values can be better characterized. In the case of pedotransfer function based mapping it is advantageous that calculation of uncertainty is much less computation intensive than it is with geostatistical methods, although it would be interesting in the future to analyze the difference between uncertainty maps calculated with the different methods specifically for soil hydraulic properties.

P12 L30-32: A discussion of methods or suggestions of alternative methods should be done in the discussion section.

A: It will be moved under Results and discussion section.

P12 L31 – P13 L2: The conclusion is the wrong place for such a detailed discussion of the methods used. A new table (Table 7) should not be introduced in the conclusion section.

A: Both text and Table 7 will be moved under Results and discussion section.

P13L2-3: A comparison with findings by other authors should be done in the discussion section. New references should not be introduced in the conclusion (e.g. Webster and Oliver (2017)).

A: Text will be moved under Results and discussion section.

Thank you again the comments and suggestions. We hope to adequately address the issues identified and look forward to any other feedback the referee may have.

———————————————————

[Figure]

THS of topsoils on TEST_CHEM set (N = 2448 )

THS of subsoils on TEST set (N = 3611 )

FC of topsoils on TEST_CHEM set (N = 2416 )

FC of subsoils on TEST set (N = 3579 )

WP of topsoils on TEST_CHEM set (N = 2459 )

WP of subsoils on TEST set (N = 3611 )

**Fig. 1.** P1

[Figure]

**Fig. 2.** P2

[Figure]

**Fig. 3.** P3

[Figure]

Hungarian Unified
National Projection
System

0  5  10 km

N
W — E
S

**Range of water
content values**

cm³ cm⁻³

| | |
|---|---|
| | < 0.12 |
| | 0.12 - 0.14 |
| | 0.14 - 0.16 |
| | 0.16 - 0.18 |
| | 0.18 - 0.2 |
| | 0.2 - 0.22 |
| | 0.22 - 0.24 |
| | 0.24 < |

**Fig. 4.** P4

---

## Author Comment (AC2) · 11 Feb 2019

General comments

This is an interesting manuscript investigating an important topic. The manuscript is well structured, but it is difficult to follow in places. Especially the methods need further clarification and details.

A: Thank you for the review and suggestion for clarification. We will improve the intelligibility of the text, especially of the methods' description. Please find our answers and proposed solutions for your specific comments hereinafter.

[Figure]

Specific comments

Q1: Lines 11-13: very long sentence

A1: We will rephrase it:

Spatial 3D information on soil hydraulic properties for areas larger than plot scale are usually derived by indirect methods, such as pedotransfer functions (PTFs) due to lacking measured information on those. PTFs describe the relationship between the desired soil hydraulic parameter and easily available soil properties based on a soil hydraulic reference dataset. Soil hydraulic properties of a catchment or region can be calculated with applying PTFs on available soil maps.

Q2: Lines 13-16: not clear

A2: Thank you for highlighting it, we will rephrase the sentence according to the followings:

Our aim was to analyse the performance and spatial distribution of soil hydraulic properties derived with (i) indirect (using PTFs) and (ii) direct (geostatistical) mapping methods. We performed the study on Balaton catchment in Hungary, where density of measured soil hydraulic data fulfils the requirements of geostatistical methods.

Q3: Abstract: in general: a lot of acronyms for an abstract. Not clear and difficult to understand what direct and indirect are.

A3: We will eliminate the following abbreviations: THS, FC, WP, MARTHA. The direct mapping method is based on geostatistical analysis, in this study we used random forest with kriging. In the indirect mapping method first we derive the prediction function of soil hydraulic properties. We will try to make it clearer with small edits in the abstract: ". . . soil hydraulic properties derived with (i) indirect (using PTFs) and (ii) direct (geostatistical) mapping methods. . . . As a direct, thus geostatistical method random forest combined with kriging (RFK) was applied . . ."

Q4: Introduction: I think a language revision is needed. The first paragraph for example is difficult to understand.

A4:. Thank you for the suggestion. Along the suggestions of the two reviewers we made a lot of corrections for the improvement of the manuscript. If necessary, we will turn to a native English speaker for further corrections and improvements.

Q5: Lines 19-22: this is not always true. Please discuss.

A5: Thank you for highlighting it. The sentence will be modified similarly to the following: "It has been shown, that often but not always the more models are combined for the prediction the more accurate the result is (Baker and Ellison, 2008; Cichota et al., 2013; Nussbaum et al., 2018; Wu et al., 2018). Although significancy of improvement is often not tested."

Q6: Section 2.2: the description of the soil hydraulic dataset is not clear. The split between test and training in particular.

A6: Thank you for highlighting it. We will rewrite the description on splitting the dataset into train and test similarly to the following: " For the construction of PTFs those samples were selected from the MARTHA dataset which have measured values of soil horizons or layers considered as dependent and independent variables. We needed two kinds of predictions: (1) for topsoils where we could include organic matter content, calcium carbonate content and pH among the predictors and (2) for subsoils without the above soil chemical parameters, because those are not available for the 30-60 and 60-90 cm soil depths on the Balaton catchment. First we randomly selected 67% of the samples from those which have data on the dependent and all the independent variables available on the catchment area to derive the PTFs. The rest 33% was used to compare the performance of the PTFs, this we called TEST_CHEM set. In the second step we needed a training (67% of data) and test set (33% of data) also for subsoil prediction for which we didn't have to apply the restriction on the soil chemical properties, therefore we could include more samples for the analysis. As test set we used
the samples of the TEST_CHEM set and further added cases to reach the 33% of the complete data appropriate for subsoil predictions. Again the left 67% was used for the training. Number of samples used to train and test the PTFs was 8157 and 12039 for THS, 8051 and 11931 for FC, 8195 and 12036 for WP, with and without soil chemical properties respectively."

We will move the above paragraph under 2.4.1 to clarify that this splitting was performed for the PTF approach.

Under 2.2 we will include information only on the soil hydraulic dataset, adding some clarification about what kind of data is included in the dataset: "For the prediction of soil hydraulic properties based on soil properties and other environmental variables we used the Hungarian Detailed Soil Hydrophysical Database (Makó et al., 2010) extended with topographical, meteorological, geological information and remotely sensed vegetation properties (Table 1), called MARTHA ver 3.0 (acronym of the Hungarian name of the dataset). MARTHA consists of 15142 soil horizons' data belonging to 3970 soil profiles. The samples in it have measured information on basic soil properties – e.g. soil depth, organic matter content, clay, silt and sand content, calcium carbonate content, pH, etc. – and also on soil hydraulic properties such as soil water retention at different matric potential values."

Q7: The results are well presented, but there are minor (and less minor) problems with typos and structure of the sentences."

A7: Thank you. We will correct the following typos related to RMSE and R2 values similarly to it:

"RMSE values calculated on the test sets for RF were between 0.042 and 0.045 cm3 cm-3 for THS, 0.039 and 0.042 cm3 cm-3 for FC, 0.035 and 0.038 cm3 cm-3 for WP, which is close to the performance of other internationally accepted PTFs (e.g. Botula et al. (2013), Román Dobarco et al. (2019), Zhang and Schaap (2017)). R2 was 0.408-0.487, 0.746-0.766 and 0.737-0.762 for THS, FC and WP respectively on test

sets in the case of RF."

Further typos and structure of the sentences will be checked by native speaker.

---

## Author Response (AR1)

**Revison of manuscript on "Mapping soil hydraulic properties using random forest based pedotransfer functions and geostatistics"**

**Content of the document**

| I. POINT BY POINT AUTHORS' RESPONSE TO THE REVIEWERS    | 1    |
|---------------------------------------------------------|------|
| 1. RESPONSE TO THE EDITOR                               | 1    |
| 2. RESPONSE TO REFEREE #1                               | 2    |
| 3. RESPONSE TO REFEREE #2                               | 9    |
| II. LIST OF ALL RELEVANT CHANGES MADE IN THE MANUSCRIPT | . 11 |
| III. MARKED-UP MANUSCRIPT VERSION                       | . 13 |

**I. POINT BY POINT AUTHORS' RESPONSE TO THE REVIEWERS**

**1. RESPONSE TO THE EDITOR**

Dear Dr. Ehret,

Thank you for the positive evaluation of our replies and highlighting those three important points which will significantly improve the clarity of the manuscript. Please find here our answers following the raised points with page and line numbering of the revised manuscript:

**Q1:** Language: Although there will be copy-editing done by HESS after the manuscript has been accepted, please have a native speaker read your manuscript to check for spelling, grammar and readability.

A1: The whole manuscript has been checked by an English language expert.

**Q2:** Conclusions: As referee #1 mentioned, a conclusions section should be short and concise. It should pick up the main questions/hypotheses of the manuscript, provide the related findings of the paper and point to deficiencies and required future work.

**A2:** We were not aware of the correct formulation of the conclusions and wrongly included discussion in that section. The text has been completely moved under Results and discussion section – to P14 L12 - P15 L16 – and we added the real conclusions there – P15 L23-28.

**Q3:** As mentioned by referee #1 (P6L20-P7L8), please make clear the difference between your two main approaches, and why you call one 'direct' and the other 'indirect'. If this is made clear from the very beginning of the paper, it will be much easier to understand. I recommend adding an illustrative flowchart explaining the sequence of steps of each method.

A3: Thank you for highlighting that the difference between the methods was not completely clear and giving the idea of the flowchart. We have added a paragraph under 2.4 section -P6 L5-13 -, in which we summarize the main steps of the two methods and added the flowchart on it as Fig. 2.

We hope we could properly address all the questions in the revised version of the article.

With regards, Authors

**2. RESPONSE TO REFEREE #1**

Dear Referee #1,

Thank you for the detailed review and suggestions which help us to improve our manuscript. We have tried to address all the comments in the revised version of the article. Below we would like to answer the questions and recommendations, following the general and special comments. **Page and line numbering refer to that of the revised manuscript with track changes.**

**General comments:**

**Q1:** The methods used to predict soil retention characteristics are adequate. However, the procedure of variable selection does not become totally clear.

**A1:** With the variable selection our aim was to exclude less important predictors. Please find detailed description on how variables have been selected under answers for the specific comments and revised description on it on P7 L22-28.

**Q2:** The manuscript is largely well structured, minor changes are suggested in the specific comments below. The conclusion is not in an appropriate form at all and should be written again.

**A2**: Thank you for calling our attention to reformat the conclusions. Information and Table 7 have been moved under results and discussion to P14 L12-P15 L16 and conclusions have been rewritten on P15 L23-28. Please find more information on it under answers for specific comments.

**Q3:** I followed the link in the manuscript but could not download the maps and pedotransfer functions.

**A3:** We are sorry that the download link of the maps (https://www.mta-taki.hu/en/kh124765/maps) did not work, something happened with the access authorization after submitting the manuscript, now the problem has been fixed. We also added a link to download the HUN-PTFs: https://www.mta-taki.hu/en/kh124765/hun\_ptfs on P16 L29-P17 L2.

**Specific comments:**

Abstract

P1L16-17: Please formulate more precise: "water content at saturation (THS), at field capacity (FC), and at the wilting point (WP)"

A: Thank you for the suggestion, information on matric potential values has been added in the abstract as well: P1 L18-19.

**Introduction**

P2 L29 – P3 L2: In this paragraph only studies are listed in which tree- based MLA algorithms worked best. Are there also studies where other methods like e.g. artificial neural networks performed best? If yes, they should also be mentioned here. I also think that tree-based methods are a very good choice in this study, but I wonder if there is really only one best approach.

A: Thank you for your comment. We have provided some papers, which used several MLAs (e.g. neural networks, cubist, gradient boosting) for mapping soils and short summary of them: P3 L4-11.

P3 L10: Do you mean soil water content at field capacity and wilting point?

A: Thank you for highlighting it, it has been clarified: P3 L22

P3 L11: How can measurements be optimized? What is meant by number of measurements? A large number of?

A: You are right, the sampling density was optimized, the sentence has been clarified accordingly: P3 L23-24.

P3 L14-21: Please provide some numbers summarizing the uncertainties found in the studies cited, so the reader can get a feeling about which order of magnitude of uncertainties can be expected when predicting soil retention data. This might also define the "internationally accepted performance of hydraulic PTFs" mentioned in the abstract (P1 L22).

A: Indeed, it is important, thank you for the idea. We have added this information in the text on P3 L33-P4 L7 and the following references:

Leenaars, J. G. B., Claessens, L., Heuvelink, G. B. M., Hengl, T., Ruiperez González, M., van Bussel, L. G. J., Guilpart, N., Yang, H. and Cassman, K. G.: Mapping rootable depth and root zone plant-available water holding capacity of the soil of sub-Saharan Africa, Geoderma, 324(February), 18–36, doi:10.1016/j.geoderma.2018.02.046, 2018.

Nguyen, P. M., Haghverdi, A., de Pue, J., Botula, Y.-D., Le, K. V., Waegeman, W. and Cornelis, W. M.: Comparison of statistical regression and data-mining techniques in estimating soil water retention of tropical delta soils, Biosyst. Eng., 153, 12–27, doi:10.1016/j.biosystemseng.2016.10.013, 2017.

P3 L23-25: The objective of the study should be clear and unambiguous. The formulation of the aim(s) should therefore always be identical when mentioned in the text (in the abstract, in the last paragraph of the introduction and in the first paragraph of the conclusion).

A: Thank you for the suggestion. We have rephrased the objective mentioned in the text to be identical. We stick to the following main aim: to analyse difference in performance and spatial patterns between soil hydraulic maps derived with indirect (using PTFs) and direct (geostatistical) mapping methods.

The possibility for a non-computation intensive method to map uncertainty of calculated soil hydraulic parameters is a possible advantage of the PTF method.

Please find changes in text on P4 L9-12.

**Materials and Methods**

P4 L12-20: The quite large number of abbreviations introduced in the manuscript unnecessarily demands the capacity of the reader. Please omit abbreviations when the term is used only a few

times (e.g. ST or PSD).

A: We decreased the number of abbreviations, MLA, OM, CaCO3 and EU-SHG are not used in the text and unnecessary ST and PSD abbreviations have been removed too (eg.: P2 L12, L24; P5 L1; P4 L27-30, etc.). The THS, FC, WP and MARTHA abbreviations have been deleted from the abstract: P1 L18-9, 23, 27.

P4 L22: The covariates are only used to predict the soil hydraulic properties. The relationships between the response and predictor variables are not analysed in the manuscript (e.g. by partial dependence plots). Please rephrase "analysis of the relationships".

A: Thank you for highlighting it, the sentence has been rephrased on P5 L7-8.

P4 L22: What does the number 173 stand for? Is it the number of available covariates?A: Yes, we have deleted it to keep the sentence simpler on P5 L8. Number of predictors is mentioned elsewhere, it might not be necessary to highlight it also here.

P4 L30 – P5 L2: I had to read the sentences several times to understand them. Please rephrase. A: We have rephrased those sentences on P6 L18-30

P4 L25 – P5 L5: The content of the paragraph is not really covered by the heading "Soil hydraulic dataset". Please adapt the heading. I also asked myself, if some information should be shifted to section 2.4.1.

A: Thank you for the suggestion. Information on how data was selected to train and test the PTFs has been moved under section 2.4.1. (P6 L18-30) The heading has been rephrased on P5 L6: Dataset to relate soil hydraulic properties and environmental information.

P5 L7: Please rephrase "most often used soil water retention values".

A: We have rephrased it: "We mapped soil water content at 0, -330 and -15,000 cm matric potential values, THS, FC and WP respectively, because these soil hydraulic properties are often required for various purposes".

P5 L7-8: Why did you map water content at -330 cm matric potential when field capacity is determined in Hungary at -300 cm?

A: Thank you for finding it, -300 was a mistyping error. It has been corrected in the text on P5 L28.

P5 L15: Why are these methods the most efficient MLAs? This is a very general statement. I am sure that many data scientist would at least partially disagree. Please rephrase. See also my comment on P2 L29 – P3 L2.

A: We have rephrased the sentence on P7 L2-4. RF and GBM are two widely used MLA, which often achieve good prediction performance on datasets that are characterized by a large number of predictors.

P5 L17: To calculate quantiles during the predictions? Quantiles of what? What is meant by "during the predictions"?

A: The text has been rephrased on P7 L4-5.

P5 L15-L19: Please add some general information about the principles of regression trees. Also an unexperienced reader should get at least an imagination how the input information is transformed

to water retention characteristics. Please also mention once the alternative names of the MLA s (e.g. boosted regression trees) to avoid confusion.

A: We have summarized the principles of regression trees before describing the difference between RF and GMB on P7 L7-15

P5 L19: ...build ensembles of models...

A: It has been corrected on P7 L7.

- P5 L19: ... the difference between GBM and RF is the way... A: It has been corrected on P7 L8-9.
- P5 L26: mtry? Seems to be an argument of an R function? A: Yes, we have clarified it in the text on P7 L18.

P6 L1: 50 independent variables out of how many? I assume that it is related to the number 173 in P4 L21. Right?

A: Yes for topsoils, in the case of subsoils it was 170, it has been clarified in the text on P7 L26-27

P6 L1: It is not really clear to me how you performed the variable selection. Especially when potential predictors are correlated it can be quite challenging to find an optimal set of predictors. Did you start with all possible predictors at once or did you try out many different combinations of predictors?

A: Thank you for highlighting it, the sentence was not properly phrased. Our aim was to reduce the number of predictors. We selected the 50-50 most important variables both in GBM and RF methods based on the five times repeated five-fold cross-validation, then concatenated the two sets of predictors. In this way less relevant predictors were excluded from the analysis. First we wanted to use the recursive feature elimination (Gregorutti et al., 2017) – with rfe function implemented in R caret package –, which would be a real optimization of input variable selection, but the RFE analysis couldn't be finished on the training set (173 variables of more than 5700 samples) due to lacking computation capacity. Then we found the possibility to at least reducing the number of predictors based on importance measure of the variables. Nussbaum et al. (2018) compared different covariate selection methods: a) based on variable importance calculated in RF model and b) stepwise recursive elimination of the least important variables. They found that both methods selected similar set of covariates. Their study was similar to our ones regarding the topic and dimension of data, therefore based on their results we reduced the number of predictors based on variable importance, which is practically the first step of the RFE analysis. In HUN-PTF method we considered the variable importance of both GBM and RF to rely on the results of two different methods. We concatenated the 50-50 most important variables, in this way depending on soil hydraulic parameter and soil depth 65-76 predictors stayed in the model. Text has been modified accordingly on P7 L22-28.

Gregorutti, B., Michel, B. and Saint-Pierre, P.: Correlation and variable importance in random forests, Stat. Comput., 27(3), 659–678, doi:10.1007/s11222-016-9646-1, 2017.

P6 L8-11: Terminology again: Is it right that "out of bag sampling" is identical to "bootstrapping"? If yes, you might also drop the term "bootstrapping" once.

A: Yes, we have added the term bootstrapping under 2.4.1. section on P7 L10 and P8 L3.

P6 L18: . . . to the median and the 5% and 95% quantiles. . .

A: Thank you, we have modified it on P8 L12-13.

P6 L20 – P7 L8: The combination of state of the art MLAs and classical geostatistical tools seems plausible and promising to me. However, Iwonder if it is correct to call it simply "direct mapping". Isn't it a combination of both: indirect (prediction with RF) and direct (kriging) mapping? Maybe I just haven t understood the essential differences between direct and indirect mapping approaches.

A: Thank you for your observation. We have added a paragraph under 2.4 section – P6 L5-13 – , in which we summarize the main steps of the two methods and added a flowchart as Fig. 2, which highlights the difference between the direct and the indirect method. The essential difference between direct and indirect mapping is the approach of the inference. In direct mapping the target soil variable is directly interpolated over the domain of interest, whereas in indirect mapping not the target variable but its components / factors / covariates are interpolated first and then these interpolated surfaces are in use to compute and map the target variable. Pásztor et al. (2017) discussed this two approaches in detail.

P6 L26: . . . Table 2 summarizes the measured. . .

A: Thank you, we have modified it on P8 L21-22

P7 L2: Here it says "most important covariates" (the result of the variable selection, right?), but in the caption of Table 1 it says (all) "available environmental covariates".

A: Thank you for highlighting it. Yes, Table 1 shows all available environmental covariates, text has been corrected on P8 L27.

P7 L11: ... with the method...

A: Thank you, we have modified it on P9 L5.

P7 L10: "...based on measured soil hydraulic properties calculated for...". How can the measured properties be calculated? Please rephrase the sentence.

A: It has been modified on P9 L4-5.

**Results and discussion**

P7 L27-28: In P6 L1 it says that most important 50 independent variables have been selected. How did you select them out of the 69-76 and 65-77 variables mentioned here?

A: We have added information on how number of variables was decreased under 2.4.1 section on P7 L27-28.

Based on both GBM and RF analysis most important 50-50 variables was selected, after concatenating those, we got 69-76 for topsoil predictions, for subsoils 65-77 stayed depending on the target variable in the case of HUN-PTF method.

P8 L9-16: This paragraph should be shifted to the Materials and Methods section.

A: The paragraph includes the result of the tuning of the model parameters before building the final model, therefore we thought to include it under the results of HUN-PTFs on P10 L11-18.

P8 L6: Why can you assume that multicollinearities are no problem at all? I assume, that many of the predictors presented in Table 1 are highly correlated. I wonder if it is even possible to estimate a

unique set of regression-tree parameters when predictors are correlated. For the same reason I could also imagine that it is not possible to determine one unique set of 50 most important independent variables.

A: Thank you to highlight it, the sentence has been revised and complemented on P10 L2-9.

In case of our analysis multicollinearity is similar in the training set and mapped area therefore it had less influence on the performance of the maps. It is true that if the HUN-PTFs would be applied in a different region, multicollinearities might influence the performance of the predictions. Dorman et al. (2013) found that prediction performance of random forest did not get worse due to high collinearity in the training dataset. The above mentioned RFE analysis would help to decrease multicollinearity (Gregorutti et al., 2017), but didn't run on our dataset due to high dimensionality. By eliminating around 100 predictors from the entire 173 based on the importance measures, we could partially decrease the multicollinearity, and improve performance of the prediction. Optimizing predictor selection could be further elaborated, but this is beyond the aim of the presented paper.

Dormann, C. F., Elith, J., Bacher, S., Buchmann, C., Carl, G., Carré, G., Marquéz, J. R. G., Gruber, B., Lafourcade, B., Leitão, P. J., Münkemüller, T., Mcclean, C., Osborne, P. E., Reineking, B., Schröder, B., Skidmore, A. K., Zurell, D. and Lautenbach, S.: Collinearity: A review of methods to deal with it and a simulation study evaluating their performance, Ecography (Cop.)., 36(1), 027–046, doi:10.1111/j.1600-0587.2012.07348.x, 2013.

P8 L22: Please compare the values listed in the text and in Table 1 once again. I am not sure if they match.

A: Thank you, you are right, we have corrected it on P10 L23-27. We have modified the  $R^2$  values as well, we listed those only for RF. In this way we keep the logic: highlighting results of the selected algorithms.

P8 L27 and many other passages in the text: Is it correct to use the term "covariate" when talking about regression trees? Tome "predictors" or "independent variables" seems more plausible.

A: Thanks for highlighting it. We used the environmental covariates as independent variables in HUN-PTF and the RF part of the RFK. For clarification we introduced the term "predictors" in text related to PTFs: under 2.1. (P5 L1), 2.4. (P6 L20; P7 L11, 15, 23, 24, 26, 28; P8 L8, 28), 3.1. (P10 L2, 32), 3.2 (P12 L7). Geostatisticians use the term "environmental covariate", therefore it might enhance the interpretability of the manuscript if also this term would be kept.

P8 L30: ... than soil related variables...

A: Thank you, it has been corrected on P11 L2.

- P9 L6-L26: Please explain in the Method section how relative importance is determined. A: Explanation on it has been added, on P7 L23-25.
- P9 L30: mtry? See also my comment on P5 L26.

A: We have written it out on P12 L7: "number of randomly selected predictors at each split" for easier understanding.

P9 L27 - P10 L13: In addition to the quality criterions presented in Table 4 it would be interesting to see scatterplots (measured versus predicted values). They sometimes give a better feeling for model performance and they also show if there are areas in the predicted data space of THS, FC and WP with very good or poor prediction performance.

A: Thank you for the suggestion we have added the scatter plots in Fig 4, which shows the 90 % prediction intervals and refer it in the text on P10 L27-30.

Figure 4. The scatter plot of the measured versus predicted water retention values with 90% prediction interval on test data sets based on random forest method. THS: saturated water content, FC: water content at field capacity, WP: water content at wilting point, TEST\_CHEM set: test dataset in which chemical soil properties are available for the predictions, TEST set: test dataset, in which chemical soil properties are not necessarily available for the predictions.

P10 L6-13: Please discuss what it is good for to add kriged values computed with a pure nugget model when the residuals of the RF predictions show no spatial structure. This way you simply add random numbers that blur your predicted mean values. I wonder, if you should leave out the whole exercise.

A: You are right in that sense kriged values computed with a pure nugget model do not give any new "information" to the RF predictions. However, kriged values with a nugget variogram add zero values to the RF predictions rather than random numbers. Thus kriging with a nugget model do not blur the predictions. We would not like to leave this exercise because it is an algorithmical decision – considering the stochastic part of the spatial variation of the given soil property – rather than a subjective decision, even if we get the same result.

P10 L10: the correlation is based on only three pairs of values. Please use a weaker formulation.

A: Thank you, we have rephrased the sentence on P12 L18-19.

P10 L33 – P11 L9 and Fig. 5: Why did you select WP in Fig 5 and why did you only show confidence intervals for HUN-PTF? It would also be interesting to see maps of THS and FK and the confidence interval from the RFK predictions.

A: We have added maps also for THS and FC, please find those as Fig 7, 8, 10. Text has been modified accordingly on P13 L10-12, 19.

Sorry for confusion, the formulation of the aim of the paper has been cleared in the entire manuscript. Calculating the confidence intervals for the RFK method is beyond the scope of this study, although it would be interesting to analyze the difference between uncertainty maps calculated with the different methods in the future, similarly as it was done by Szatmári and Pásztor (2018) for soil organic carbon stock in Hungary. According to it, quantile regression forest (Meinshausen, 2006) based uncertainty quantification outperforms most of the prediction techniques used in digital soil mapping. Furthermore, they have pointed out that bootstrapping based uncertainty quantification for RFK is quite time consuming, as well as it requires massive storage and computing capacity. The ranger package - with which we derived the HUN-PTFs - includes implementation of quantile regression forest (Meinshausen, 2006) for the calculations of the prediction intervals.

Information related to bootstrapping based uncertainty quantification for RFK has been added on P15 L8-12.

Meinshausen, N.: Quantile Regression Forests, J. Mach. Learn. Res., 7, 983–999, 2006.

P11 L12: . . . we have not differentiated uncertainty of. ..

A: Thank you, it has been corrected on P13 L23-24.

**Conclusion**

P12 L1 – P13 L3: The conclusion has poor quality and should be written again. A conclusion should just consist of one or two paragraphs where the most important results are summarized and the

most important conclusions are drawn. A concise take home massage can be formulated. In the following just some examples of aspects are listed that are wrong placed the conclusion of the manuscript: P12 L17- 20: Such general methodological aspects are not the take home message of the study.

A: We were not aware of the correct formulation of the conclusions and wrongly included discussion in that section. The text has been completely moved under Results and discussion section to P14 L12-P15 L16, and we have added the real conclusions on P15 L23-28. The take home message is the following:

Based on results of six out of nine soil hydraulic maps there is no significant difference in performance between values derived using pedotransfer function and geostatistical method on the Balaton catchment area. The benefit of maps computed with random forest and kriging is that locally extreme values can be characterized better. In the case of pedotransfer function based mapping it is advantageous that calculation of uncertainty is much less computation intensive than it is with geostatistical methods, although it would be interesting in the future to analyse the difference between uncertainty maps calculated with the different methods specifically for soil hydraulic properties.

P12 L30-32: A discussion of methods or suggestions of alternative methods should be done in the discussion section.

A: It has been moved under Results and discussion section to P15 L4-6.

P12 L31 – P13 L2: The conclusion is the wrong place for such a detailed discussion of the methods used. A new table (Table 7) should not be introduced in the conclusion section.

A: Both text and Table 7 have been moved under Results and discussion section to P15 L13-15.

P13L2-3: A comparison with findings by other authors should be done in the discussion section. New references should not be introduced in the conclusion (e.g. Webster and Oliver (2017)).

A: Text has been moved under Results and discussion section to P15 L15-16.

Thank you again the comments and suggestions. We hope that we could adequately address the issues identified and look forward to any other feedback the referee may have.

With regards, Authors

**3. RESPONSE TO REFEREE #2**

Dear Referee #2,

Thank you for the review and suggestions for clarification, which helps to improve the quality of the manuscript. We hope that we could address all the raised questions and comments in the revised version of the manuscript. Please find our answers for the questions and recommendations, following the general and specific comments. **Page and line numbering refer to that of the revised manuscript with track changes**.

**General comments**

**Q1:** This is an interesting manuscript investigating an important topic. The manuscript is well structured, but it is difficult to follow in places. Especially the methods need further clarification and details.

**A1:** Thank you for the review and suggestion for clarification. The entire manuscript has been checked by a specialized language expert to improve intelligibility of the text. The followings have been modified/included related to description of the methods:

- information and flowchart (Fig.2) has been added to clarify difference between direct and indirect methods on P6 L5-13,

- the number of abbreviations has been decreased, MLA, OM, CaCO3, EU-SHG, ST and PSD have been removed from the text (eg.: P2 L12, L24; P5 L1; P4 L27-30, etc.)

- data partition to train and test the pedotransfer functions has been rephrased on P6 L18-30,

- information on how data was selected to train and test the PTFs has been moved under section 2.4.1. and rephrased (P6 L18-30),

- the heading has been rephrased on P5 L6: Dataset to relate soil hydraulic properties and environmental information, and only information about the soil hydraulic dataset has been kept there (P5 L7-13),

- the principles of regression trees have been summarized before describing the difference between RF and GMB on P7 L7-15,

- meaning of mtry has been clarified on P7 L18 and P12 L7,

- information on how number of variables was decreased has been rephrased on P7 L22-28 to increase clarity,

- the sentence about multicollinarity has been revised and complemented on P10 L2-9.

**Specific comments**

Q1: Lines 11-13: very long sentence

**A1:** Thank you for highlighting it. We have rephrased the first part of the abstract on P1 L10-15.

Q2: Lines 13-16: not clear

**A2:** We have structured the sentence on P1 L15-16 and rephrased the sentences on P1 L15-18.

**Q3:** Abstract: in general: a lot of acronyms for an abstract. Not clear and difficult to understand what direct and indirect are.

**A3:** We have eliminated the following abbreviations from the abstract: THS, FC, WP, MARTHA. In the direct method we used the geostatistical approach to spatially inference measured soil hydraulic data collected in profiles of the catchment through modelling its relationship with environmental covariates. In indirect mapping PTFs were derived first to describe relationships between soil hydraulic properties and easily available soil and other environmental parameters. The PTF predictions were then spatially implemented on the environmental covariates clipped for the catchment area of Lake Balaton. We have added some small edits in the abstract to clarify it:

- on P1 L15-16: "... soil hydraulic properties derived from (i) indirect (using PTFs) and (ii) direct (geostatistical) mapping methods" and

- on P1 L22-23: "As a direct, thus geostatistical method random forest combined with kriging (RFK) was applied ..."

**Q4:** Introduction: I think a language revision is needed. The first paragraph for example is difficult to understand.

**A4:** Thank you for the suggestion. Along the suggestions of the two reviewers we made a lot of corrections for the improvement of the manuscript. The language of the manuscript has been edited by a language expert.

**Q5:** Lines 19-22: this is not always true. Please discuss.

**A5:** Thank you for highlighting it. The sentence has been rephrased on P2 L21-23.

**Q6:** Section 2.2: the description of the soil hydraulic dataset is not clear. The split between test and training in particular.

**A6:** Thank you for highlighting it. We have rephrased the description on splitting the dataset into train and test on P6 L18-30. We have moved that paragraph under section 2.4.1 to clarify that this splitting was performed for the HUN-PTF approach.

Under section 2.2 (P5 L6-13) we have only kept information about the soil hydraulic dataset, and added some clarification about what kind of data is included in the dataset on P5 L11-13, and rephrased the title of the section (P5 L6).

**Q7:** The results are well presented, but there are minor (and less minor) problems with typos and structure of the sentences.

**A7:** Thank you. We have corrected the typos related to RMSE and R2 values on P10 L23-26. Further typos and structure of the sentences has been corrected by a language expert.

Thank you for the review. We hope that we could clarify text related to the methods and improve the intelligibility of the entire manuscript. We are looking forward the feedback about the revised manuscript.

With regards, Authors

**II. LIST OF ALL RELEVANT CHANGES MADE IN THE MANUSCRIPT**

Page and line numbering refer to that of the revised manuscript with track changes.

The following changes have been made in the manuscript:

- the English has been edited in the entire manuscript;
- the following abbreviations have been removed from the text of the manuscript: MLA, OM, CaCO3, EU-SHG, ST and PSD;
- for clarification we introduced the term "predictors" in text related to PTFs: under 2.1. (P5 L1),
   2.4. (P6 L20; P7 L11, 15, 23, 24, 26, 28; P8 L8, 28), 3.1. (P10 L2, 32), 3.2 (P12 L7)
- P1 L4: the first author will use her maiden name: Brigitta Szabó;
- abstract has been rephrased to clarify the text and decrease abbreviations: P1 L10-19, L21, L23, L27;
- introduction:
  - P2 L21-23: text has been rephrased and a sentence has been added for clarification;

- P3 L1-2: tested machine learning algorithms are listed,
- P3 L4-11: short summary of papers has been added, which used several machine learning algorithms (e.g. neural networks, cubist, gradient boosting) for mapping soils;
- P3 L22-24: two sentence have been clarified;
- P3 L33-P4 L7: published numbers on the magnitude of uncertainties related to the prediction of soil water retention have been added;
- P4 L9-12: aim of the study has been clarified;
- materials and methods:
  - P5 L6: title of section has been modified to better describe its content;
  - P5 L7-8: sentence has been clarified;
  - P5 L11-13: information about the properties included in the MARTHA dataset has been added;
  - P5 L14-24: text has been moved under 2.4.1 to P6 L15-30 to clarify that the dataset was divided in a certain way only in the case of the HUN-PTF method, the text has been rephrased to increase intelligibility;
  - P6 L4: title has been modified;
  - P6 L5-13: a summary about the main steps of the two methods and a flowchart on it as Fig. 2 has been added;
  - P7 L1: sentence has been rephrased;
  - P7 L4-5: sentence on computing uncertainty has been clarified;
  - P7 L7-15: summary on the principles of regression trees has been added;
  - P7 L10, P8 L3: the term bootstrap sample is used for clarification;
  - P7 L18: meaning of argument mtry has been clarified;
  - P7 L21-28: description about the selection of predictors has been rephrased;
  - P8 L12-14: terms have been corrected and sentence has been rephrased;
  - P8 L27: clarification has been added;
  - P9 L4-5: sentence has been clarified;
  - P9 L19: reference of R software has been added through Mendeley's Word Plug In;
- results and discussion:
  - P9 L28-P10 L10: text related to variable selection and multicollinearity has been completed;
  - P10 L24-26: typos have been removed;
  - P10 L27-30: scatterplot as Fig. 4 and its description has been added;
  - P12 L7: mtry has been replaced with its description;
  - P12 L18-19: the sentence has been rephrased;
  - P13 L12-14: maps of THS and FC has been added on separate figures: Fig. 7, 8, and added to Fig. 10;
  - P14 L11-P15 L16: a new section has been introduced, text and Table 7 from previous version of the conclusions (P15 L29-P6 L27) have been moved here;
  - P15 L8-12: information related to bootstrapping based uncertainty quantification for RFK has been added;
- conclusions
  - P15 L23-28: conclusions has been added, text related to the discussion (P15 L19-23 and P15 L29-P16 L27) has been removed as mentioned before;
- data availability:
  - P16 L29-P17 L2: link to download HUN-PTFs has been added;
- references: new reference has been added with Mendeley's Word Plug In therefore not highlighted with track changes – on
  - P18 L3-4,
  - P18 L31-34,
  - P19 L13-14,
  - P19 L17-19,

- P20 L16-18,
- P20 L31-33,
- P21 L11-13,
- P21 L34-P22 L1
- tables:
  - P26 L3-4: information has been added on test sets;
  - P26 L5: information on soil type has been clarified;
- figures: numbering of figures has been revised due to adding four more figures,
  - P33: flowchart (Fig. 2) has been added, which describes direct and indirect method;
  - P35: scatterplot (Fig. 4) has been added, which shows measured vs predicted values with 90% prediction intervals;
  - P37 L4-5: sentence has been clarified;
  - P38: map of THS of 0-30 cm soil depth has been added (Fig. 7);
  - P39: map of FC of 0-30 cm soil depth has been added (Fig. 8);
  - P40-41: map of WP of 0-30 cm soil depth has been replaced (Fig. 9) and its caption has been clarified;
  - P42-43: figure has been replaced by map also including THS and FC (Fig. 10).

**III. MARKED-UP MANUSCRIPT VERSION**

Please find revised marked-up manuscript on the following pages.

**Mapping soil hydraulic properties using random forest based pedotransfer functions and geostatistics**

Brigitta SzabóTóth1,2, Gábor Szatmári1, Katalin Takács1, Annamária Laborczi1, András Makó1, Kálmán 5 Rajkai1, László Pásztor1

[revised manuscript text omitted]

and vegetation listed in Table 1 was used as predictors and environmental covariates for the elaboration of PTFs as well as for and direct mapping accordingly.

Topographical parameters were calculated with SAGA GIS tools (Conrad et al., 2015) based on the digital elevation model. For the mapping of soil hydraulic properties all covariates were harmonized, projected to the Hungarian Uniform National Projection system, rasterized if necessary and resampled to 100 m resolution.

**2.2 Soil hydraulic dDataset to relate soil hydraulic properties and environmental information**

For the analysis of the relationship betweenprediction of soil hydraulic properties and based on soil and other environmental covariates variables (173) 
[revised manuscript text omitted]

| Pred          | icted soil    | Colocted | Train set**    |                                             |      | TEST set       |                                             | TEST_CHEM set |                |                                             |      |
|---------------|---------------|----------|----------------|---------------------------------------------|------|----------------|---------------------------------------------|---------------|----------------|---------------------------------------------|------|
| hydr:
prop | aulic
erty | method*  | $\mathbb{R}^2$ | RMSE
(cm 3 cm -3 ) | Ν    | $\mathbb{R}^2$ | RMSE
(cm 3 cm -3 ) | Ν             | $\mathbb{R}^2$ | RMSE
(cm 3 cm -3 ) | Ν    |
| THS           |               | GBM      | 0.453          | 0.052                                       | 5709 |                |                                             | -             | 0.484          | 0.042                                       | 2448 |
|               | topson        | RF       | 0.488          | 0.041                                       | 5709 |                |                                             | -             | 0.487          | 0.042                                       | 2448 |
|               | 1 1           | GBM      | 0.429          | 0.045                                       | 8428 | 0.41           | 3 0.045                                     | 3611          | 0.400          | 0.046                                       | 2448 |
|               | subsoil       | RF       | 0.480          | 0.043                                       | 8428 | 0.429          | 0.045                                       | 3611          | 0.408          | 0.045                                       | 2448 |
| FC            | 1             | GBM      | 0.714          | 0.043                                       | 5635 |                |                                             | -             | 0.770          | 0.039                                       | 2416 |
|               | topson        | RF       | 0.736          | 0.041                                       | 5635 |                |                                             | -             | 0.766          | 0.039                                       | 2416 |
|               | 1 1           | GBM      | 0.738          | 0.044                                       | 8352 | 0.73           | 0.042                                       | 3579          | 0.751          | 0.040                                       | 2416 |
|               | subsoil       | RF       | 0.756          | 0.042                                       | 8352 | 0.74           | 5 0.042                                     | 3579          | 0.759          | 0.040                                       | 2416 |
| WP            | 1             | GBM      | 0.722          | 0.038                                       | 5736 |                |                                             | -             | 0.739          | 0.037                                       | 2459 |
|               | topson        | RF       | 0.736          | 0.037                                       | 5736 |                |                                             | -             | 0.762          | 0.035                                       | 2459 |
|               | 1 1           | GBM      | 0.717          | 0.041                                       | 8425 | 0.71           | 5 0.039                                     | 3611          | 0.711          | 0.038                                       | 2459 |
|               | subsoil       | RF       | 0.747          | 0.039                                       | 8425 | 0.73           | 0.038                                       | 3611          | 0.744          | 0.036                                       | 2459 |

5 \* Input parameters included in all analysis for topsoils: Hungarian-soil type according to Hungarian classification system, sand (50–2000 μm), silt (2–50 μm) and clay content (<2 μm) (100 g g-1), mean depth (cm) and information on topography, vegetation, meteorology and parent material listed in Table 1. For subsoils organic matter content (100 g g-1); pH in water and calcium carbonate content (100 g g-1) were included as well.
\*\* Prediction error calculated on training is based on out of bag error in case of RF and 5-fold cross-validation in case of GBM method.

| Predicted soil          | Donth | Random forest  |                                               |              | Variogram |       |        |
|-------------------------|-------|----------------|-----------------------------------------------|--------------|-----------|-------|--------|
| hydraulic
properties | (cm)  | $\mathbb{R}^2$ | RMSE
(cm 3 cm -3 ) N | Partial sill | Туре      | Range | Nugget |
| THS                     | 0-30  | 0.403          | 0.055 324                                     | 0            | "Nug"     | -     | 32.552 |
|                         | 30-60 | 0.251          | 0.055 321                                     | 11.037       | "Exp"     | 1531  | 18.357 |
|                         | 60-90 | 0.189          | 0.060 315                                     | 14.150       | "Exp"     | 8211  | 27.067 |
| FC                      | 0-30  | 0.562          | 0.053 324                                     | 0            | "Nug"     | -     | 29.895 |
|                         | 30-60 | 0.532          | 0.056 321                                     | 0            | "Nug"     | -     | 26.539 |
|                         | 60-90 | 0.478          | 0.063 315                                     | 0            | "Nug"     | -     | 32.356 |
| WP                      | 0-30  | 0.463          | 0.052 324                                     | 0            | "Nug"     | -     | 23.689 |
|                         | 30-60 | 0.474          | 0.051 321                                     | 0            | "Nug"     | -     | 22.655 |
|                         | 60-90 | 0.466          | 0.056 315                                     | 32.718       | "Sph"     | 2149  | 0      |

Table 4. Performance of random forest method and parameters of the fitted variogram models during the geostatistical mapping approach.

| Predicted soil hydraulic
property | Depth    | Method  | Ν   | RMSE
(cm 3 cm -3 ) | SS mse | Sign. difference* |
|--------------------------------------|----------|---------|-----|---------------------------------------------|-------------------|-------------------|
| THS                                  | 0-30 cm  | RFK     | 324 | 0.056                                       | 0.382             | b                 |
|                                      |          | HUN-PTF | 350 | 0.067                                       | 0.118             | b                 |
|                                      |          | EU-SHG  | 348 | 0.070                                       | 0.041             | а                 |
|                                      | 30-60 cm | RFK     | 321 | 0.060                                       | 0.119             | а                 |
|                                      |          | HUN-PTF | 345 | 0.058                                       | 0.150             | b                 |
|                                      |          | EU-SHG  | 343 | 0.063                                       | -0.004            | а                 |
|                                      | 60-90 cm | RFK     | 315 | 0.063                                       | 0.112             | b                 |
|                                      |          | HUN-PTF | 337 | 0.060                                       | 0.171             | с                 |
|                                      |          | EU-SHG  | 335 | 0.071                                       | -0.149            | а                 |
| FC                                   | 0-30 cm  | RFK     | 324 | 0.053                                       | 0.547             | b                 |
|                                      |          | HUN-PTF | 350 | 0.067                                       | 0.265             | b                 |
|                                      |          | EU-SHG  | 348 | 0.076                                       | 0.070             | а                 |
|                                      | 30-60 cm | RFK     | 321 | 0.057                                       | 0.515             | b                 |
|                                      |          | HUN-PTF | 345 | 0.069                                       | 0.278             | b                 |
|                                      |          | EU-SHG  | 343 | 0.084                                       | -0.069            | а                 |
|                                      | 60-90 cm | RFK     | 315 | 0.062                                       | 0.485             | b                 |
|                                      |          | HUN-PTF | 337 | 0.074                                       | 0.232             | b                 |
|                                      |          | EU-SHG  | 335 | 0.095                                       | -0.243            | а                 |
| WP                                   | 0-30 cm  | RFK     | 324 | 0.052                                       | 0.453             | b                 |
|                                      |          | HUN-PTF | 349 | 0.062                                       | 0.244             | ab                |
|                                      |          | EU-SHG  | 347 | 0.071                                       | -0.038            | а                 |
|                                      | 30-60 cm | RFK     | 321 | 0.052                                       | 0.467             | b                 |
|                                      |          | HUN-PTF | 344 | 0.065                                       | 0.152             | b                 |
|                                      |          | EU-SHG  | 342 | 0.074                                       | -0.112            | a                 |
|                                      | 60-90 cm | RFK     | 315 | 0.057                                       | 0.443             | с                 |
|                                      |          | HUN-PTF | 335 | 0.067                                       | 0.208             | b                 |
|                                      |          | EU-SHG  | 333 | 0.076                                       | -0.026            | а                 |

Table 5. Performance of soil hydraulic maps derived by random forest and kriging method (RFK), Hungarian pedotransfer functions (HUN-PTF) and from EU-SoilHydroGrids 250m dataset (EU-SHG) on the Balaton catchment. RMSE: root mean square error, SSmse: mean square error skill score.

\*Different letters indicate significant differences at 0.05 level between the accuracy of the methods based on squared error,

5 e.g. performance indicated with letter c is significantly better than the one noted with letter b and a.

Table 6. Proportion of mapped area having smaller than 0.025, 0.025-0.050, 0.05-0.100 and bigger than 0.10 cm3 cm-3 absolute difference between predicted soil hydraulic values derived by geostatistical method (RFK) and applying pedotransfer functions on local soil and environmental covariates (HUN-PTF).

| Absolute difference between                         |            | % 0   | % of mapped area |    |  |  |
|-----------------------------------------------------|------------|-------|------------------|----|--|--|
| RFK and HUN-PTF (cm 3 cm -3 ) | Deptn (cm) | THS F | C V              | VP |  |  |
| 0-0.025                                             | 0-30       | 76    | 80               | 71 |  |  |
|                                                     | 30-60      | 86    | 77               | 65 |  |  |
|                                                     | 60-90      | 75    | 72               | 71 |  |  |
| 0.025-0.050                                         | 0-30       | 21    | 17               | 25 |  |  |
|                                                     | 30-60      | 10    | 21               | 26 |  |  |
|                                                     | 60-90      | 21    | 22               | 24 |  |  |
| 0.050-0.100                                         | 0-30       | 3     | 3                | 4  |  |  |
|                                                     | 30-60      | 4     | 2                | 9  |  |  |
|                                                     | 60-90      | 4     | 6                | 5  |  |  |
| 0.100 <                                             | 0-30       | 0     | 0                | 0  |  |  |
|                                                     | 30-60      | 0     | 0                | 0  |  |  |
|                                                     | 60-90      | 0     | 0                | 0  |  |  |

Table 7. Differences between pedotransfer function based (PTF) and geostatistical (RFK) mapping methods based on calculating saturated water content, field capacity and wilting point for the Balaton catchment.

| Aspects of                                                                                      | Differences between the soil hydraulic mapping methods                                                                                                                                                                                                                                                                                                                                                                                                                                                                                                                                                          |                                                                                                                                                                                                                                                                                                                                                                                                                                                                                                                                                                          |  |  |  |  |
|-------------------------------------------------------------------------------------------------|-----------------------------------------------------------------------------------------------------------------------------------------------------------------------------------------------------------------------------------------------------------------------------------------------------------------------------------------------------------------------------------------------------------------------------------------------------------------------------------------------------------------------------------------------------------------------------------------------------------------|--------------------------------------------------------------------------------------------------------------------------------------------------------------------------------------------------------------------------------------------------------------------------------------------------------------------------------------------------------------------------------------------------------------------------------------------------------------------------------------------------------------------------------------------------------------------------|--|--|--|--|
| mapping                                                                                         | PTF – indirect method                                                                                                                                                                                                                                                                                                                                                                                                                                                                                                                                                                                           | RFK – direct method                                                                                                                                                                                                                                                                                                                                                                                                                                                                                                                                                      |  |  |  |  |
| Main steps of
mapping                                                                        | 1. derive PTFs on available soil hydraulic dataset or use an appropriate PTF available from the literature, 2. apply PTFs on available environmental covariates                                                                                                                                                                                                                                                                                                                                                                                                                                                 | <ol> <li>harmonize soil profile dataset available
for the mapping based on required soil
depth, 2. predict deterministic component,</li> <li>calculate the residuals, estimate their
variograms, krige them, 4. add kriged
residuals to the deterministic component</li> </ol>                                                                                                                                                                                                                                                                           |  |  |  |  |
| Dataset used to
describe
relationship
between soil
hydraulic data and
covariates |  <li>any soil hydraulic dataset which is
hydropedologically similar to the area for
which soil hydraulic maps are required</li> <li>advantages: mapping can be applied even if
no soil hydraulic data is available for the
study area; available PTF also can also be
used</li> <li>disadvantages: a soil hydraulic dataset is
needed which has to be similar to data of the
study site from soil hydropedological point
of view; or if PTF is already available the
soil hydrological dataset used to train the
PTF has to be similar to the study site</li>  |  <li>soil hydraulic data available for the catchment</li> <li>advantages: soil hydraulic data is characteristic for the study site, locally extreme values can be better characterized</li> <li>disadvantages: density of measured soil hydraulic properties available for the study site might not satisfy the needs for mapping; further to the soil property, which is mapped, measured data of soil properties used in the prediction of the deterministic component (e.g. particle size distribution, OM organic matter content) is required as well</li>  |  |  |  |  |
| Inclusion of soil
depth                                                                      |  <li>can be included as independent variable</li> <li>advantages: measured soil hydraulic
properties are related to measured soil
properties; soil hydraulic properties at any
depth can be calculated</li> <li>disadvantages: certain depths can be
underrepresented in the training dataset
which might increase prediction uncertainty</li>                                                                                                                                                                                                                                     |  <li>in 2D kriging soil data (chemical, physical, hydraulic) is first harmonized in training dataset by splining to derive data for fix depth</li> <li>disadvantages: measured soil properties are splined therefore calculated soil hydraulic properties are related to calculated soil properties, thus map relationship between them is derived from interpolated (namely splined) values</li>                                                                                                                                                               |  |  |  |  |
| Spatial inference                                                                               |  <li>this method relies on the interpolation
included in the input layers used for the
mapping, thus the mapping is indirect</li> <li>advantage: no further geostatistical
analysis is needed to provide 3D information</li>                                                                                                                                                                                                                                                                                                                                                               |  <li>directly the soil hydraulic properties are interpolated</li> <li>advantage: uncertainty of input layers is decreased due to adding the kriged residuals to the predicted values</li>                                                                                                                                                                                                                                                                                                                                                                       |  |  |  |  |

l

|                               | - disadvantage: uncertainty of input layers
increase uncertainty of predicted soil
hydraulic properties                                                                                                                                                                                                                                                                                                                                                       |                                                                                                                                                                                                                                                                                                                                                                                                |
|-------------------------------|---------------------------------------------------------------------------------------------------------------------------------------------------------------------------------------------------------------------------------------------------------------------------------------------------------------------------------------------------------------------------------------------------------------------------------------------------------------------|------------------------------------------------------------------------------------------------------------------------------------------------------------------------------------------------------------------------------------------------------------------------------------------------------------------------------------------------------------------------------------------------|
| Information on
uncertainty |  <li>interpreted as the uncertainty of the PTFs</li> <li>advantage: can be easily computed for PTFs</li> <li>disadvantages: not location specific, but depends on the input parameter combination, uncertainty of input layers has to be added to the uncertainty of PTFs to provide information on the uncertainty of soil hydraulic maps, uncertainty of input environmental covariates is hardly definable if e.g. 60-70 of them are used for the</li>  |  <li>can be derived with e.g. bootstrapping</li> <li>advantages: location specific; the uncertainty accounts for both the unexplained stochastic variation and the uncertainty in estimating the deterministic model</li> <li>disadvantages: computationally demanding; require massive storage capacity; uncertainty of input layers has to be added to the uncertainty of RFK</li>  |
|                               | mapping                                                                                                                                                                                                                                                                                                                                                                                                                                                             |                                                                                                                                                                                                                                                                                                                                                                                                |

---

## Author Response (AR2)

**2nd revison of manuscript on "Mapping soil hydraulic properties using random forest based pedotransfer functions and geostatistics"**

**Content of the document**

**I. POINT BY POINT AUTHORS' RESPONSE TO THE REVIEWERS**

**1. RESPONSE TO THE EDITOR**

Dear Dr. Ehret,

Thank you for the positive evaluation of our manuscript. Please find here our answers following the raised point with page and line numbering of the revised manuscript with track changes:

**Q1:** I agree with the referee suggestion to both harmonize your objectives throughout the manuscript and make them more specific. Please revise your manuscript accordingly, which should not take much time.

    **A1:** We have harmonized the objectives in the abstract (P1 L14-16), introduction (P4 L5), results and discussion (P13 L1-3) and conclusions (P14, L29-30) according to the reviewer suggestion.

We hope that the harmonization of the objectives has been addressed well with the modification.

With regards,
Authors

**2. RESPONSE TO REFEREE #1**

Dear Referee #1,

Thank you for dedicating your time for the thorough review and constructive suggestions which helps us improving the quality of our manuscript. Below we would like to answer the question and recommendation following the comments. **Page and line numbering refer to that of the revised manuscript with track changes.**

**General comments:**

**Q1:** The authors have thoroughly answered to all the comments I raised about the first version of the manuscript. In almost all cases, they have adopted the manuscript concerning my comments and questions. To my mind, they have substantially improved the quality of the manuscript. I recommend the work to be published in HESS after minor revision.

>   **A1:** Thank you for helping us improving clarity and quality of our manuscript and the positive evaluation.

**Specific comment:**

**Q2:** My only comment relates to the formulation of the objectives. In my opinion, it is the most important sentence of the manuscript. I still recommend using the same formulation in the Abstract, the Introduction and the beginning of the Conclusion.

In the Abstract it says: "Our aim was to analyse the performance and spatial distribution of soil hydraulic properties derived from (i) indirect (using PTFs) and (ii) direct (geostatistical) mapping methods". Here the phrase "performance of soil hydraulic properties" is not clear. I guess that the performance of the methods to derive the soil hydraulic properties is meant, right?

In the last paragraph of the Introduction it says: "Our aim was to analyse how different mapping methods could be applied to derive maps of soil hydraulic properties, such as water content at saturation (THS), field capacity (FC) and wilting point (WP) on the Balaton catchment area in Hungary". Here "different mapping methods" is too vague for an objective formulation when the specific method are of central importance.

Maybe you can find another more concise and unambiguous formulation of the objectives.

>   **A2:** Thank you for keeping attention on it and suggesting solution. You are right, performance is related to the method, the text on the objectives has been modified accordingly:
>
>   **-** Abstract P1 L14-16: sentence has been simplified and clarified,
>
>   **-** Introduction P4 L5: sentence has been specified,
>
>   **-** Results and discussion P13 L1-3: rephrased to increase clarity,
>
>   **-** Conclusions P14 L29-30: cleared to increase readability.

**Further remark**

**Q3:** I further recommend clarifying the copyright of the provided soil hydraulic maps of the Balaton catchment and the pedotransfer functions. You can define how your data may be used by others. This might also help them to adequately acknowledge your work. I recommend assigning a creative commons licence and indicating it with the cc label. You can find all required information at https://creativecommons.org/. This is just a recommendation and no reviewer claim.

>   **A3:** Thank you for the suggestion, we are checking how to apply https://creativecommons.org/ and how it is different compared to https://opendatacommons.org/licenses/odbl/.

We hope that we could adequately harmonize the objectives.

With regards,
Authors

**II. LIST OF ALL RELEVANT CHANGES MADE IN THE MANUSCRIPT**

Page and line numbering refer to that of the revised manuscript with track changes.

The following changes have been made in the manuscript:
- P1 L4: affiliation of András Makó has been corrected,
- P1 L9: e-mail address of contact author has been changed,
- P1 L14-16: sentence has been simplified and clarified,
- P1 L16: "a study" was replaced with "the study",
- P4 L5: sentence has been specified,
- P11 L32: unnecessary enter has been deleted,
- P13 L1-3: rephrased to increase clarity,
- P14 L29-30: cleared to increase readability,
- P21 L34-P22 L1: unnecessary enters have been deleted,
- Table 1: font size has been set, table has been formatted,
- Table 2: font size has been set,
- Table 7: font size has been set.

**III. MARKED-UP MANUSCRIPT VERSION**

Please find revised marked-up manuscript on the following pages.

[revised manuscript text omitted]

Hungarian Unified
National Projection
System

0    5    10
▬▬▬▭▭ km

N
W ⟨⊕⟩ E
S

**Range of water
content values**
cm³ cm⁻³

☐ < 0.12
☐ 0.12 - 0.14
☐ 0.14 - 0.16
☐ 0.16 - 0.18
☐ 0.18 - 0.2
☐ 0.2 - 0.22
☐ 0.22 - 0.24
☐ 0.24 <

**Figure 10. Differences between possible lower 5 % and upper 95 % water content at saturation (a), field capacity (b) and wilting point (c) in 0-30 cm soil depth for a section of the Balaton catchment.**